# Differentiable hierarchical and surrogate gradient search for spiking neural networks

**Kaiwei Che**[1,2 *], **Luziwei Leng**[1,2 * ✉], **Kaixuan Zhang**[1,2], **Jianguo Zhang**[1],
**Max Q.-H. Meng**[1], **Jie Cheng**[2], **Qinghai Guo**[2], **Jiangxing Liao**[2]
[1] Southern University of Science and Technology, China
[2] ACS Lab, Huawei Technologies, Shenzhen, China

## Abstract

Spiking neural network (SNN) has been viewed as a potential candidate for the next generation of artificial intelligence with appealing characteristics such as sparse computation and inherent temporal dynamics. By adopting architectures of deep artificial neural networks (ANNs), SNNs are achieving competitive performances in benchmark tasks such as image classification. However, successful architectures of ANNs are not necessary ideal for SNN and when tasks become more diverse effective architectural variations could be critical. To this end, we develop a spike-based differentiable hierarchical search (SpikeDHS) framework, where spike-based computation is realized on both the cell and the layer level search space. Based on this framework, we find effective SNN architectures under limited computation cost. During the training of SNN, a suboptimal surrogate gradient function could lead to poor approximations of true gradients, making the network enter certain local minima. To address this problem, we extend the differential approach to surrogate gradient search where the SG function is efficiently optimized locally. Our models achieve state-of-the-art performances on classification of CIFAR10/100 and ImageNet with accuracy of 95.50%, 76.25% and 68.64%. On event-based deep stereo, our method finds optimal layer variation and surpasses the accuracy of specially designed ANNs meanwhile with $26\times$ lower energy cost (6.7mJ), demonstrating the advantage of SNN in processing highly sparse and dynamic signals. Codes are available at `https://github.com/Huawei-BIC/SpikeDHS`.

## 1   Introduction

Inspired from biological neural networks, spiking neural network (SNN) [41] has been viewed as a potential candidate for the next generation of artificial intelligence, with appealing characteristics such as asynchronous computation, sparse activation and inherent temporal dynamics. However, the training of deep SNNs is challenging due to the binary spike which is incompatible with gradient-based backpropagation. To solve this problem, various surrogate gradient (SG) methods were proposed [5, 65, 48] where soft relaxed functions were used to approximate the original discontinuous gradient. Based on these methods, SNNs have achieved high level performances on benchmark image classification tasks such as CIFAR and ImageNet [58, 66, 74, 54, 76]. However, the accuracy of SNN often drops when directly adopting ANN architectures , such as ResNet [21] and VGG networks [59]. With recent improved SNN training methods [35, 15, 13] the performance gap is decreasing but remains. This gap is more evident in tasks where the network architecture requires more variation, such as dense image prediction [81, 20, 25].

Directly inheriting sophisticated ANN architectures might not be ideal for SNN and there is *a lack of study* of optimal architectures for spiking neurons given a particular task. Intuitively, topology

---

[*]Equal Contributions. [✉]Corresponding author. lengluziwei@huawei.com

36th Conference on Neural Information Processing Systems (NeurIPS 2022).

in cortex should offer us some inspiration. Neuroscience has studied macro structures such as columnar organization [44] and micro dynamics such as interaction between lateral and feedback connections [36]. However, on the network level the relation between connections of neural circuits and functionalities remains largely unknown. In this work, we develop a spike-based differentiable hierarchical search (SpikeDHS) framework in order to find optimal task-specific network architectures under limited computation cost. Based on traditional differentiable architecture search (DARTS) framework [40], we redesign the search space and information flow in the principle of spike-based computation. For both the cell and the layer search space, we study how to realize this principle towards a fully spiking network.SG provides an approximation of the non-existing gradient in SNN and its selection is not unique. To explore optimal SG functions for the training of SNN, we propose a differentiable SG search method to efficiently adapt the function locally, and demonstrate its efficiency on both static and event-based benchmark machine learning tasks. In summary, our contributions are following:

- We develop a differentiable hierarchical search framework for spiking neurons, realizing spike-based computation on both the cell and the layer level search space, based on which optimal SNN architectures can found under limited computation cost.

- To improve gradient approximation of deep SNNs, we propose a differentiable SG search method to efficiently optimize SG functions locally, which is easy to scale and also effective for binary networks.

- Extensive experiments show that our methods outperform SNNs based on sophisticated ANN architectures on image classification of CIFAR10, CIFAR100 and ImageNet datasets.

- On event-based deep stereo task, to the best of our knowledge we show the first time SNN surpasses specially designed ANNs on the Multi Vehicle Stereo Event Camera (MVSEC) [80] dataset in terms of accuracy, network sparsity and computation cost, demonstrating its advantage in processing highly sparse and dynamic signals with extremely low power and latency.

## 2 Related work

### 2.1 Architecture search

Designing high performance network architectures for specific tasks often requires expert experience and trial-and-error experiments. Neural architecture search (NAS) [14] aims to automate this manual process and has recently achieved highly competitive performance in tasks such as image classification [82, 83, 38, 55, 52], object detection [83, 8, 64, 19] and semantic segmentation [39, 75, 49, 37], etc. However, searching over a discrete set of candidate architectures often results in a massive number of potential combinations, leading to explosive computation cost. The recently proposed differentiable architecture search (DARTS) method [40] and its variations [67, 7, 10] address this problem using a continuous relaxation of the search space which enables learning a set of architecture coefficients by gradient descent, and has achieved competitive performances with the state-of-the-art using orders of magnitude fewer computation resources [40, 39, 9]. Recently, [46] studied pooling operations for downsampling in SNNs and applied NAS to reduce the the overall number of spikes. [26] applied NAS to improve SNN initialization and explore backward connections. However, both works only searched for different SNN cells or combinations of them under fixed network backbone and their application is limited to image classification.

### 2.2 Training of SNN

The success of deep ANNs in solving benchmark machine learning tasks has motivated efforts to make SNNs realize similar capabilities, either based on bio-inspired mechanisms [51, 47, 33, 45, 32, 27] or approximating ANN learning algorithms [5, 65, 48, 3]. Currently, two approaches have demonstrated their efficiency, showing the ability to solve hard problems at similar levels as their artificial counterparts, namely, ANN-to-SNN conversion [57, 6, 34] and directly training SNNs with SG [58, 66, 74, 35, 15, 13]. The theoretical soundness of the SG approach for training binary activation networks has been studied and justified [4, 69]. Experiments show that the training of SNN is robust to the shape of SG function as long as it meets certain criteria, such as the overall scale [71]. [20] shows that a suitable SG function is critical when the SNN goes deeper. [35] further improved

the performance of SNN by optimizing the width (or temperature) of a continuous SG function, through the guidance of an approximated gradient measured with finite difference gradient (FDG). However, a suitable hyperparameter setting of FDG is largely empirical. In terms of computation, it iterates over each element of the weight and is sequential across layers, making a global application of this method computationally impractical.

## 2.3 Event-based task with SNN

Inspired by biological retina, event camera [18] captures instantaneous changes of pixel intensity at microsecond resolution. Compared to traditional frame-based cameras, it covers a higher dynamical range (120dB) and offers a low power solution for vision tasks in high-speed scenario. Further combining it with neuromorphic processors [50, 56, 11, 42, 17, 30] can create a fully neuromorphic system, realizing extremely low power and low latency sensing. However, learning from highly sparse and asynchronous events is challenging. Given its inherited asynchronous dynamics, SNN is an ideal candidate for such task and recently a number of works have applied it for event-based problems such as classification [29, 34], tracking [68], detection [24], semantic segmentation [25] and optical flow estimation [31, 20], etc. Multi-view event-based deep stereo solves the problem of 3D scene reconstruction based on pixel differences of the same physical point from event streams obtained by multiple views. Given the problem's complexity, it has been addressed by specially designed deep ANNs. Several works use additional information such as camera motion to produce sparse depth maps [79, 78]. Estimating dense disparity images from sparse event inputs is more challenging. A recent work [62] addresses this problem by an event queue method which encodes events into event images through 3D convolution, followed by an hourglass network to estimate disparity. Based on [62], [1] further enhances local contours of the estimated disparity using image reconstruction.[43] creates feature pyramids with multi-scale correlation learned from a cycle of gray-scale images and event inputs. Inspired from biological neuron models, [73] developed discrete time convolution to encode events with temporal dynamic feature maps and proposed a dual-path encoder with spatially adaptive modulation to strengthen events representation. A very recent work [53] applies SNN to this problem with a handcrafted U-net structure. Nevertheless, compared with ANN models using geometric volumes its performance is suboptimal.

## 3 Method

### 3.1 Preliminary

We adopt the iterative leaky integrate-and-fire (LIF) neuron model [66] described by

$$u^{t,n} = \tau u^{t-1,n}(1 - y^{t-1,n}) + I^{t,n} \tag{1}$$

where superscripts $n$ and $t$ denote layer index and time step, respectively. $\tau$ is the membrane time constant, $u$ is the membrane potential, $y$ denotes the spike output and $I$ denotes the synaptic input with $I^{t,n} = \sum_j w_j y_j^{t,n-1}$ where $w$ is the weight. The neuron will fire a spike $y^{t,n} = 1$ when $u^{t,n}$ exceeds a threshold $V_{th}$, otherwise $y^{t,n} = 0$. In this work, we set $\tau = 0.2$ and $V_{th} = 0.5$. Given loss $L$, using chain rule the weight update of SNN can be expressed as:

$$\frac{\partial L}{\partial w} = \sum_t \frac{\partial L}{\partial y^t} \frac{\partial y^t}{\partial u^t} \frac{\partial u^t}{\partial I^t} \frac{\partial I^t}{\partial w} \tag{2}$$

where $\frac{\partial y^t}{\partial u^t}$ is the gradient of the spiking function, which is zero everywhere except at $u = V_{th}$. The SG approach uses continuous functions to approximate the real gradients, such as rectangular [76], triangular [2], Superspike [70], ArcTan [16] and exponential curves [58]. We adopt Dspike function from [35]:$\text{Dspike}(x) = a \cdot \tanh(b(x - c)) + d$, which can cover a large range of smoothness by changing the temperature parameter $b$, with $\text{Dspike}(x) = 1$ or 0 for $x > 1$ or $x < 0$. We set $c = V_{th} = 0.5$ and determine $a$ and $d$ by setting $\text{Dspike}(0) = 0$, $\text{Dspike}(1) = 1$.

### 3.2 Differentiable hierarchical search for SNN

#### 3.2.1 Cell level search

Similar to traditional DARTS [40], a cell is defined as a repeated and searchable unit, which is a directed acyclic graph with $N$ nodes, $\{x_i\}_N$, as depicted in Fig. 1. Each cell receives input from two

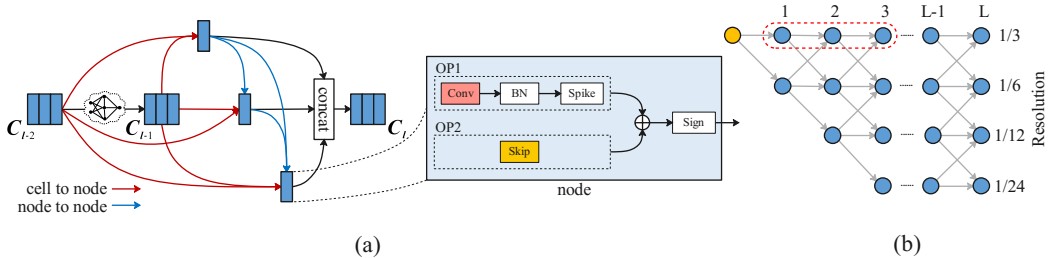

**Figure 1:** Hierarchical search space of SpikeDHS. (a) Cell level search space, with 3 nodes as an example. Within one node, operations are summed (2 operations here) and filtered by a sign function. BN denotes batch normalization. (b) Layer search space for event-based deep stereo.

previous cells and forms its output by concatenating all outputs of its nodes. In SpikeDHS, each node is a spiking neuron described by:

$$x_j = f(\sum_{i<j} o^{(i,j)}(x_i)) \tag{3}$$

where $f$ is a sign function (or a spiking neuron) taking the sum of all operations as input, $o^{(i,j)}$ is the operation associated with the directed edge connecting node $i$ and $j$. During search, each edge is represented by a weighted average of candidate operations, the information flow connecting node $i$ and node $j$ becomes:

$$\bar{o}^{(i,j)}(x) = \sum_{o \in O^{(i,j)}} \frac{\exp(\alpha_o^{(i,j)})}{\sum_{o \in O^{(i,j)}} \exp(\alpha_o^{(i,j)})} o(x) \tag{4}$$

where $O^{(i,j)}$ denotes the operation space on edge $(i,j)$ and $\alpha_o^{(i,j)}$ is the weight of operation $o$, which is a trainable continuous variable. At the end of search, a discrete architecture is selected by replacing each mixed operation $\bar{o}^{(i,j)}$ with the most likely operation $o^{(i,j)} = \max_{o \in O^{(i,j)}} \alpha_o^{(i,j)}$. For spiking neurons, we can either mix the operation at the spike activation or at the input of the membrane potential, i.e. $y = \bar{o}$ or $I = \bar{o}$. The former allows a search over operations with different SG functions, while the latter transfers more accurate learning signals for $\alpha$ and leads to a concise node with less spiking filters (see supplement material for deduction). We choose the former for search phase and apply the latter particularly for SG search during retraining when operation type freeze (Section 3.4). Specifically, in the former choice, we distinguish between operations with the same forward computation but with different SG functions in candidate operation space. In addition, in original DARTS, a prepossessing step is required on operations between nodes and cells of previous layers in order to align the dimension of feature maps. We merge this step with subsequent candidate operations to reduce model complexity and improve inference speed.

### 3.2.2 Layer level search

Task specific knowledge has been proved to be helpful in speeding up the search process and improving network performance. For classification task, we adopt a fixed downsampling structure (Fig. 2) as in [40], with normal cells and reduction cells searched separately. For dense image prediction where high resolution output is needed and network architecture requires more variation, we implement differentiable search on the layer level as proposed in [39]. A set of scalars $\{\beta\}$ are trained to weight different potential layer resolutions and they are updated together with $\alpha$. By the end of search, an optimal structure is decoded from a pre-defined $L$-layer trellis, as shown in Fig. 1. For upsampling layers, we use nearest interpolation to maintain binary feature maps. For the stereo matching task, following volumetric approaches [23, 72, 77] we construct a feature volume which embeds geometric knowledge of the binocular input, and search cell structure separately for the feature and matching subnetworks, similar to [9]. At the end of both subnetworks, the output of the last cell is upsampled to the initial resolution of the trellis using spike-based activation. We use batch normalization and ANN stem layers in the search phase. In the retraining phase, the ANN stem layers are converted to SNN by replacing the Relu function with spike function and retraining the

weights from scratch. The reason for using Relu activation during search is to ensure more stable updating of the supernet, since deep SNNs may suffer from gradient vanish problem when the SG functions is not chosen appropriately. In extended experiments we replace the Relu function in stem layers with Dspike function under appropriate hyperparameters, e.g $b = 3$, and the search process is also stable. Parameters of batch normalization are converted into convolution weights and biases after retraining [22, 57], leading to full spike-based network for inference. We provide more details for spike-based layer search space in the supplement.

### 3.3 Loss function, estimator and optimization

For classification task, we use an auxiliary loss as in [40], with weight 0.4. For event-based stereo matching, our network produces a matching cost tensor $C$ of size $\frac{d_{max}}{2} \times h \times w$, based on which we estimate a disparity map $\hat{D}$ using a sub-pixel estimator [61].

$$\hat{D} = \sum_d D(d) \operatorname*{softmin}_{d:|\hat{d}-d|<\delta}(C_{d,y,x}) \text{, with } \hat{d} = \operatorname*{argmin}_d(C_{d,y,x}) \tag{5}$$

where $\delta = 2$ is an estimator support and $D(d) = 2d$ is a disparity corresponding to index $d$ in the matching cost tensor. We use a sub-pixel cross entropy loss [61] to train the network, which is described by:

$$L(\Theta) = \frac{1}{wh} \sum_{y,x} \sum_d \operatorname{Laplace}(D(d)|\mu = D_{y,x}^{GT}, b) \cdot \log(\operatorname*{softmin}_d(C_{d,y,x})) \tag{6}$$

where $\operatorname{Laplace}(D|\mu = D_{y,x}^{GT}, b)$ is a discretized Laplace distribution with the mean equal to the ground truth disparity $\mu = D_{y,x}^{GT}$ and diversity $b = 2$. Following bi-level optimization [40], we update weight and architecture parameters $\{\alpha, \beta\}$ alternately based on two disjoint training sets $A$ and $B$:

- Update network weights $\mathbf{w}$ by $\nabla_{\mathbf{w}} L(\mathbf{w}, \alpha, \beta)$ on $A$
- Update architecture parameters $\alpha$ and $\beta$ by $\nabla_{\alpha,\beta} L(\mathbf{w}, \alpha, \beta)$ on $B$

We use first-order approximation to speed up the search process. After search, we decode the discrete cell structure by retaining two strongest afferent edges for each node. As to network structure, we decode it by finding the maximum probability path between different layers.

### 3.4 Differentiable surrogate gradient search

A recent work [20] shows that a suitable SG function is critical when the SNN goes deeper, and [35] demonstrates that by optimizing the width (or temperature) of the SG function the performance of SNN can be improved. Continuous relaxation through gradient descent is an efficient approach to explore diverse operations on the same path, inspired from this idea, we propose a differentiable surrogate gradient search (DGS) method to parallelly optimize local SGs for SNN.

In the retraining phase, with certain epoch intervals, we associate each operation path with $N$ candidate SG functions, $\{g_i\}_N$, based on which we update the weight (or $N$ copies of the weight) separately, leading to $\{w_{g_i}\}_N$. These weights are then combined to form a mixed operation weighted by a set of factors $\{\alpha_{g_i}\}_N$ through a softmax function, described as:

$$\hat{I} = \sum_{i=1}^N \frac{\exp(\alpha_{g_i})}{\sum_{j=1}^N \exp(\alpha_{g_j})} I_i \text{, with } I_i = w_{g_i} x \tag{7}$$

We then update $\{\alpha_{g_i}\}_N$ through the loss of the mixed operation output. This process is repeated for multiple batches and finally we update the original SG to $\{g_i | i = \operatorname{argmax}_i \langle \alpha_i \rangle\}$, with $\langle \cdot \rangle$ denoting the average over batches. Note that $w_{g_i}$ can be obtained either by repeatedly calculate for each $g_i$, or directly estimating from the gradient of the original SG, $\nabla_{g,w} L$, if $\{g_i\}_N$ are linear to $g$. The psuedo code of the algorithm is summarized in Algorithm. 1. The intuition behind DGS is that the updated value of $\alpha_{g_i}$ indicates the contribution of $w_{g_i}$ in decreasing the loss. So $\{g_i | i = \operatorname{argmax}_{g_i} \langle \alpha_i \rangle\}$, which leads to the best updated weight, could be the most suitable SG function for the original local weight. Note that the difference between DGS and the SG search in Section 3.2.1 is that the former aims to optimize SG function for the local weight, while the latter is essentially a search of different operation types.

**Algorithm 1:** Differentiable surrogate gradient search (DGS)

**Input:** Training dataset, training epoch $E$, training iteration $I$, DGS iteration $I_g$, SG function $g$, candidate SG functions $\{g_i\}_N$, SG weighting factors $\{\alpha_{g_i}\}_N$ and their initializing value $\epsilon$, epoch interval for DGS $e_D$

1 **for** *all e = 1, 2, ..., E-th epoch* **do**
2     **for** *all i = 1, 2, ..., I-iteration* **do**
3        Collect training data and labels, update weights $w$ based on SG function $g$;
4     **if** $e/e_D = $ Int. **then**
5        **for** *all j =1, 2, ..., $I_g$-iteration* **do**
6           Initialize $\{\alpha_{g_i}\}_N$ to $\epsilon$, update weights $w$ with $\nabla_{g_i,w}L$ based on $\{g_i\}_N$, obtain associated weights $\{w_{g_i}\}_N$;
7           Combine $\{w_{g_i}\}_N$ to form mixed operation, update $\{\alpha_{g_i}\}_N$ with $\nabla_{\alpha_{g_i},\{w_{g_i}\}_N}L$.
8        Update $g$ to $\{g_i|i = \mathrm{argmax}_{g_i}\langle\alpha_i\rangle\}$
9 **return** trained network.

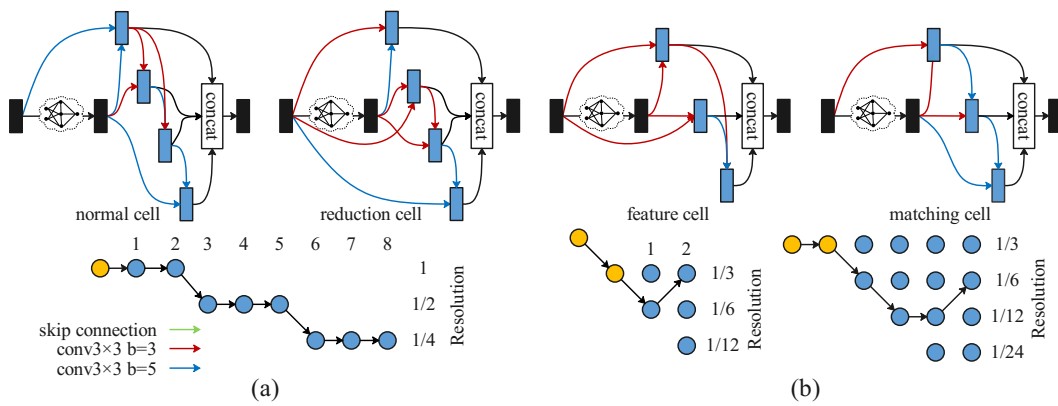

**Figure 2:** (a) Architecture for classification: SpikeDHS-CLA and (b) event-based stereo: SpikeDHS-Stereo.

## 4 Experiments

### 4.1 Classification

The CIFAR10 and CIFAR100 datasets [28] have 50K/10K training/testing RGB images with a spatial resolution of $32 \times 32$. The ImageNet dataset [12] contains more than 1250k training images and 50k test images. We apply SpikeDHS on CIFAR10 and then retrain on target datasets including CIFAR10, CIFAR100 and ImageNet. For ImageNet, we use a larger variant of the searched network with one more stem layer and two more cells. We use standard pre-processing and augmentation for training as in [21]. The test image is directly centered cropped to $224 \times 224$. More details about the model architecture and training are provided in the supplement material.

#### 4.1.1 Architecture search and retrain

In the search phase, the training set is equally split into two subsets for bi-level optimazation. For retraining, the standard training/testing split is used. We use 4 nodes (n4) within one cell and a limited number of candidate operations to reduce search time, which are {conv3×3 with $g(b = 3)$, conv3×3 with $g(b = 5)$, skip connection}, with $g = $ Dspike. For the sign function we use $g(b = 3)$. For the network architecture, we adopt an 8 layer fixed downsampling architecture proposed in [40]. The search phase takes 50 epochs with mini-batch size 50, the first 15 epochs are used to warm up convolution weights. We use SGD optimizer with momentum 0.9 and a learning rate of 0.025. The architecture search takes about 1.4 GPU day on a single NVIDIA Tesla V100 (32G) GPU. We term the searched architecture SpikeDHS-CLA and plot it in Fig. 2. After search, we retrain the model on target datasets with channel expansion for 100 epochs with mini-batch size

50 for CIFAR and 160 for ImageNet, with cosine learning rate 0.025. We use SGD optimizer with weight decay $3e^{-4}$ and momentum 0.9. The DGS method is applied to the first stem layer (s1), we set $I_g = 100$, $\epsilon = 0.001$, $e_D = 5$, $g = $ Dspike with $\{g_i\}_N$ having equal temperature interval: $\{g(b - \frac{(N-1)}{2}\Delta b), ..., g(b + \frac{(N-1)}{2}\Delta b)\}$ where $\Delta b = 0.2$ and $N = 5$. In extending experiments, we slightly increase the output channel of the first stem layer and use 3 nodes (n3) within a cell. In addition, we apply DGS to the first node of the 5th cell (c5).

### 4.1.2 Results

The results are summarized in Table 1 and the values of other models are obtained from literature. On CIFAR datasets, SpikeDHS-CLA with DGS achieves the highest accuracy comparing with other directly trained SNNs under similar model capacity. Note that both Dspike [35] and TET [13] use advanced training algorithms while plain SpikeDHS-CLA models are trained with fixed SG function. Our model also outperforms the recent SNN works with NAS in terms of accuracy, inference steps and model size[†]. As a reference, we run the original ANN DARTS network with the same architecture and it achieves 95.88% accuracy on CIFAR10. On ImageNet, our models surpass the ResNet-34-large model with much smaller model capacity and rivals VGG-16 with less than half of its size. We conjecture that the large number of jump connections both within and across cells potentially help gradient propagation through deep layers, which improves training of the model.

Table 1: Comparison on image classification. [D]: DGS method. NoP: Number of parameters.

| Dataset | Methods | Architecture | NoP | T | Accuracy[%] |
|---|---|---|---|---|---|
| CIFAR10 | [74]TSSL-BP | CIFARNet | - | 5 | 91.41 |
| | [54]Diet-SNN | ResNet-20 | - | 10 | 92.54 |
| | [76]STBP-tdBN | ResNet-19 | 13M | 6 | 93.16 |
| | [35]Dspike | ResNet-18 | 11M | 6 | $94.25 \pm 0.07$ |
| | [13]TET | ResNet-19 | 13M | 6 | $94.50 \pm 0.07$ |
| | [35]Dspike | ResNet-18 | 11M | 6 | $94.25 \pm 0.07$ |
| | [26]SNASNet | SNASNet-Bw | - | 8 | $94.12 \pm 0.25$ |
| | [46]AutoSNN | AutoSNN (C=128) | 21M | 8 | 93.15 |
| | **SpikeDHS** | SpikeDHS-CLA (n4) | 12M | 6 | $\mathbf{94.34 \pm 0.06}$ |
| | | SpikeDHS-CLA (n3) | 14M | 6 | $\mathbf{95.35 \pm 0.05}$ |
| | **SpikeDHS**[D] | SpikeDHS-CLA (n4s1) | 12M | 6 | $\mathbf{94.68 \pm 0.05}$ |
| | | SpikeDHS-CLA (n3s1) | 14M | 6 | $\mathbf{95.36 \pm 0.01}$ |
| | | SpikeDHS-CLA (n3c5) | 14M | 6 | $\mathbf{95.50 \pm 0.03}$ |
| CIFAR100 | [54]Diet-SNN | ResNet-20 | - | 5 | 64.07 |
| | [76]STBP-tdBN | ResNet-19 | 13M | 6 | $71.12 \pm 0.57$ |
| | [35]Dspike | ResNet-18 | 11M | 6 | $74.24 \pm 0.10$ |
| | [13]TET | ResNet-19 | 13M | 6 | $74.72 \pm 0.28$ |
| | [26]SNASNet | SNASNet-Bw | - | 5 | $73.04 \pm 0.36$ |
| | [46]AutoSNN | AutoSNN (C=64) | 5M | 8 | 69.16 |
| | **SpikeDHS** | SpikeDHS-CLA (n4) | 12M | 6 | $\mathbf{75.70 \pm 0.14}$ |
| | | SpikeDHS-CLA (n3) | 14M | 6 | $\mathbf{76.15 \pm 0.20}$ |
| | **SpikeDHS**[D] | SpikeDHS-CLA (n4s1) | 12M | 6 | $\mathbf{76.03 \pm 0.20}$ |
| | | SpikeDHS-CLA (n3s1) | 14M | 6 | $\mathbf{76.25 \pm 0.10}$ |
| ImageNet | [66]STBP-tdBN | ResNet-34 | 22M | 6 | 63.72 |
| | [66]STBP-tdBN | ResNet-34-large | 86M | 6 | 67.05 |
| | [54]Diet-SNN | VGG-16 | 138M | 5 | 69.00 |
| | **SpikeDHS** | SpikeDHS-CLA-large | 58M | 6 | $\mathbf{67.96}$ |
| | **SpikeDHS**[D] | SpikeDHS-CLA-large | 58M | 6 | $\mathbf{68.64}$ |

### 4.2 Event-based deep stereo

We further apply our method to event-based deep stereo matching on the widely used benchmark MVSEC dataset [80]. The dataset contains depth information recorded by LIDAR sensors and event streams collected from a pair of *Davis346* cameras, with synchronized 20 Hz gray scale images at

---

[†]The size of SNASNet is not given.

346×260 resolution. We split and preprocess the Indoor Flying dataset from the MVSEC following the same setting as [62, 1, 79]. We use the mean depth error (MDE), one-pixel-accuracy (1PA), median depth error, and mean disparity error as evaluation metrics. Learning from highly sparse raw events is challenging and a prepossessing step is often required to encode events. We use stacking based on time (SBT) [63] which merges events into temporally neighboring frames. During training, we use multiple consecutive stacks as one input, with an equal number of consecutive ground truth disparities as one label. More details can be found in the supplement material.

### 4.2.1 Architecture search and retrain

We use 3 nodes within one cell. For candidate operations, we use: {conv3×3 with $g(b = 3)$, conv3×3 with $g(b = 5)$, skip connection} for the feature net and {conv3×3×3 with $g(b = 3)$, conv3×3×3 with $g(b = 5)$, skip connection} for the matching net, with $g = \text{Dspike}$. For the layer search space, we adopt a four-level trellis with downsampling rates of {3,2,2,2}. The number of layers is set to 2 and 4 for the feature and matching net, respectively. In addition, two stem layers are applied in front of both subnetworks to reduce input spatial resolution and increase channel number. We search the architecture for 12 epochs with batch size 1. The first 3 epochs are used to initiate the weight of the supernet. The rest 9 epochs are applied with bi-level optimization. We use SGD optimizer with momentum 0.9 and a learning rate of 0.002. The architecture search takes about 0.4 GPU day on a single NVIDIA Tesla V100 (32G) GPU. We term the searched architecture as SpikeDHS-Stereo and plot it in Fig. 2. Extensive random seed experiments show that the architectures are gradually optimized during the search phase. After search, we retrain the model with channel expansion for 200 epochs with mini-batch size 2. We use Adam optimizer with initial learning rate 0.001 and momentum $(0.9, 0.999)$, with learning rate decaying half at $[50, 100, 150]$ epochs. The DGS method is applied to the first stem layer of the feature net with the same setup as in classification. In extending experiments, we also retrain the network with operation mixed at the membrane potential (MM).

### 4.2.2 Results

We compare our method with other event-based stereo matching approaches (Section 2.3) on dense disparity estimation. The results are summarized in Table 2, values of other models are obtained from literature. Among SNN methods, SpikeDHS significantly outperforms StereoSpike, exhibiting the effectiveness of the searched architecture. Among events-only approaches, SpikeDHS even surpasses ANN-based specially designed DDES network in all criteria with only one-third of its number of parameters. The performance of SpikeDHS-Stereo is further improved with DGS. As shown in Fig. 4a-b, the temperature of the SG function is constantly optimized, which avoids vanishing gradient compared to training with fixed SG function, leading to more stable training of SNN. The MM approach also improves the performance of the network. A qualitative comparison of estimated disparities is shown in Fig. 3. It can be seen that SpikeDHS with DGS predicts better disparities compares to other models, especially in local edges.

**Streaming inference:** In real-world scenario, events are generated consecutively by the sensor with flexible lengths. To test the real-time applicability of our model, we fed the entire test split continuously into the model, which evolves an equal length of steps and estimates sequential disparities. The inference speed of our model achieves 44 FPS (26 FPS for the DDES model, see supplement material for measurement details) while achieving similar accuracy as in standard testing (Fig. 3), where the model always receives a fixed length of events.

### 4.2.3 Ablation study

The inherent temporal dynamics of SNN is assumed to be helpful for it to learn temporal correlation of the data. To investigate this property, we fix the membrane time constant $\tau$ to 0 to create binary neuron (BN) and perform architecture search and retrain. As shown in Table 2, the SpikeDHS-BN model performs worse than SpikeDHS, which shows the benefit of temporal dynamics in this task. In addition, we evaluate different SG functions including Superspike [70], Triangle [2], Arctan [16] SG functions with fixed hyperparameters during training and varying hyperparameters with DGS. The results demonstrate the robustness of DGS for different SG functions as it consistently improves network performance, as shown in Table 3. More details are provided in the supplement.

**Table 2:** We denote the best and second best results in bold and underscore. EO denotes events-only method. EITNet requires gray scale images for training but not for inference. Symbols meaning: -, unavailability of the value; (·), streaming tests; $^D$, training with DGS; *, estimated value; NoP, Number of parameters.

| Method | EO | NoP | MDE [cm] ↓ | | Median depth error [cm] ↓ | | Mean disparity error [pix] ↓ | | 1PA [%] ↑ | |
|---|---|---|---|---|---|---|---|---|---|---|
| | | | split1 | split3 | split1 | split3 | split1 | split3 | split1 | split3 |
| EIS [43] | ✗ | - | **13.7** | 22.4 | - | - | - | - | 89.0 | 88.1 |
| EITNet [1] | ✓ | >16M* | 14.2 | 19.4 | **5.9** | 10.4 | 0.55 | 0.75 | **92.1** | **89.6** |
| DDES [62] | ✓ | 2.33M | 16.7 | 27.8 | 6.8 | 14.7 | 0.59 | 0.94 | 89.4 | 74.8 |
| StereoSpike [53] | ✓ | - | 18.5 | 25.4 | - | - | - | - | - | - |
| SpikeDHS-BN | ✓ | 0.87M | 17.5 | 20.3 | 7.1 | 10.8 | 0.59 | 0.74 | 88.7 | 88.3 |
| SpikeDHS-BN$^D$ | ✓ | 0.87M | 17.0 | 19.8 | 6.8 | **10.2** | 0.58 | 0.73 | 89.3 | 88.5 |
| **SpikeDHS** | ✓ | 0.87M | 16.5(16.5) | 19.4(19.5) | 6.5(6.5) | 10.6(10.6) | 0.57(0.57) | 0.73(0.74) | 90.1(90.2) | 88.5(88.4) |
| **SpikeDHS$^D$** | ✓ | 0.87M | **15.9**(15.8) | **19.1**(19.3) | 6.3(6.3) | 10.4(10.5) | **0.54**(0.54) | **0.72**(0.74) | 90.7(90.8) | 88.9(88.8) |
| **SpikeDHS (MM)** | ✓ | 0.87M | 15.7(15.7) | - | 6.3(6.3) | - | 0.55(0.54) | - | 91.0(91.1) | - |
| **SpikeDHS$^D$ (MM)** | ✓ | 0.87M | 15.4(15.4) | - | 6.0(6.0) | - | **0.54**(0.54) | - | 91.3(91.4) | - |

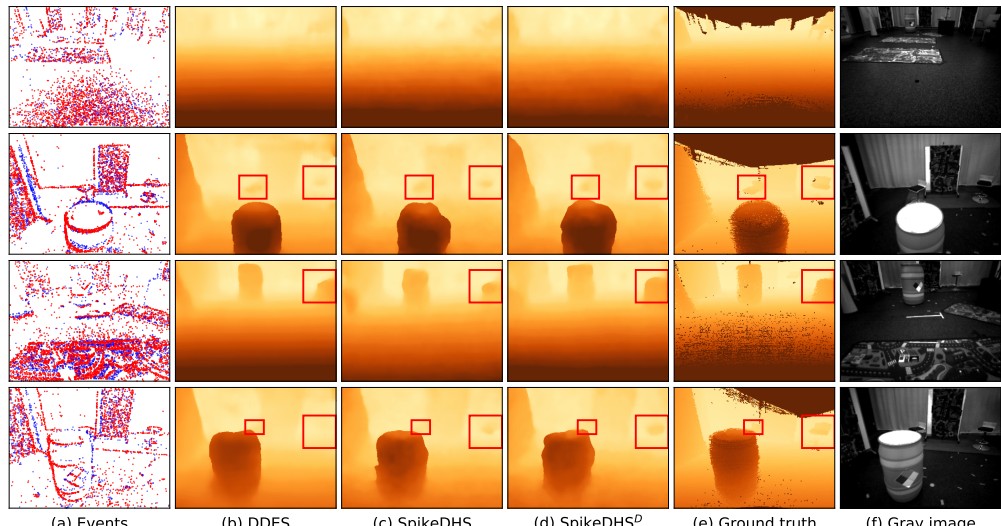

| (a) Events | (b) DDES | (c) SpikeDHS | (d) SpikeDHS$^D$ | (e) Ground truth | (f) Gray image |
|---|---|---|---|---|---|

**Figure 3:** Qualitative comparison on MVSEC. Disparity maps of different methods are on same frames, see supplement material for details.

**Table 3:** Different SG functions for event-based stereo task on split 1 w/ and w/o DGS.

| SG function | 1PA w/ DGS (w/o DGS) [%] ↑ |
|---|---|
| Triangle [2] | 91.3 (90.9) |
| Arctan [16] | 90.1 (89.3) |
| Superspike [70] | 89.7 (89.6) |

## 5 Sparsity and energy cost

The dense computation of deep ANNs came at a significant energy cost. In contrast, SNN performs sparse computing and multiplication-free inference. As Fig. 5 shows, our models show an overall sparse activity with different degrees across layers. For event-based stereo, we plot network activity along with the density of events for a short duration (Fig. 4c-d). The activity of the first stem layer in the feature net highly correlates with the density of the event streams. This relation is weakened with the increase of layer depth. As activity propagates, the sparsity of matching layers drops to create dense disparities. The operation number of ANN is collected by a public available PyTorch package [60]. Following [35], the addition count of SNN is calculated by $s * T * A$, where $s$ is the mean sparsity, $T$ is the time step and $A$ is the addition number. The operation number and

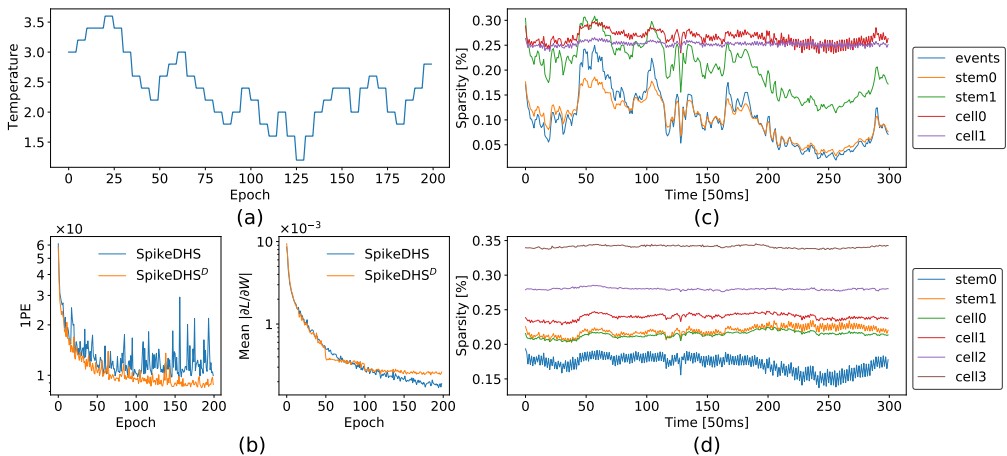

**Figure 4:** DGS training process and sparsity of SpikeDHS-Stereo. (a): Evolution of SG temperatures during DGS training. (b): 1PE (left) and first stem weight gradient (right) of DGS training and normal training. (c) and (d): Sparsity of the feature net and the matching net.

energy cost is in Table 3. We can see that SpikeDHS-Stereo has much lower operation number than DDES. Note that due to streaming inference, SNN realizes a natural usage of its temporal dynamics with $T = 1$. We measure the energy consumption following [54]. In $45\text{nm}$ CMOS technology, the addition operation in SNN costs $0.9\text{pJ}$ while the multiply-accumulate (MAC) operation in ANN consumes $4.6\text{pJ}$. SpikeDHS-Stereo costs only $6.7\text{mJ}$ for a single forward, with $26\times$ lower energy than DDES.

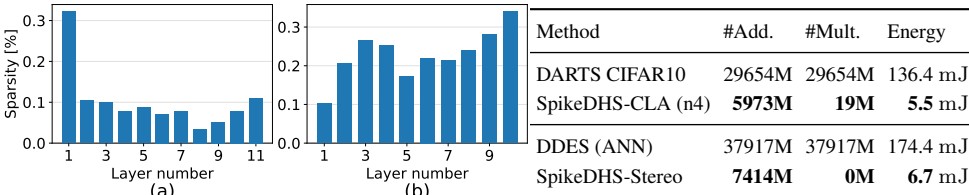

| Method | #Add. | #Mult. | Energy |
|---|---|---|---|
| DARTS CIFAR10 | 29654M | 29654M | 136.4 mJ |
| SpikeDHS-CLA (n4) | **5973M** | **19M** | **5.5** mJ |
| DDES (ANN) | 37917M | 37917M | 174.4 mJ |
| SpikeDHS-Stereo | **7414M** | **0M** | **6.7** mJ |

**Figure 5 & Table 3:** Left: Network sparsity for (a) classification and (b) event-based stereo. Right: The operation number and energy cost.

## 6  Discussion

In this paper, we develop the SpikeDHS framework which finds optimal architectures for SNNs. On image classification and event-based deep stereo tasks, our models outperform previous SNNs with ANN or handcrafted architectures. The latter task demonstrates the advantage of SNNs in processing highly sparse and dynamic event streams with extremely low power. One potential reason of the high performance of our architecture could be the large number of jump connections both within and across cells, which helps gradient propagation. Our method can be applied to other tasks where network requires more architectural variations such as object detection and semantic segmentation. The DGS algorithm efficiently optimizes SGs locally, which is of general usage in improving training of deep SNNs or binary networks. Currently we only apply DGS on a fixed layer, future works could develop a dynamical algorithm which applies it on optimal layers. In addition, we only use limited candidate operations, other types of biological plausible connections such as recurrent and feedback connections, as well as excitatory/inhibitory synapses could be considered in the future.

## Acknowledgements

This work is supported by the Science and Technology Innovation 2030-Major Project (Brain Science and Brain-Like Intelligence Technology) under Grant 2022ZD0208700.

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
