# Supplemental Material:
# Differentiable hierarchical and surrogate gradient search for spiking neural networks

**Kaiwei Che**[1,2 *], **Luziwei Leng**[1,2 * ✉], **Kaixuan Zhang**[1,2], **Jianguo Zhang**[1],
**Max Q.-H. Meng**[1], **Jie Cheng**[2], **Qinghai Guo**[2], **Jiangxing Liao**[2]
[1] Southern University of Science and Technology, China
[2] ACS Lab, Huawei Technologies, Shenzhen, China

## A   Cell search space

### A.1   Mixed operation

For spiking neurons, we can either mix the operation at the spike activation or at the input of the membrane potential. The former allows a search over operations with different SG functions, while the latter transfers more accurate learning signals for $\alpha$ and leads to a concise node with fewer spiking filters, as we show in the following.

For mixed operation at the spike function, $\bar{y} = \bar{o}$ and $y = o$. The mixed operation is then:

$$\bar{y} = \sum_k^K \frac{e^{\alpha_k}}{Z} y_k \tag{1}$$

where $Z = \sum_k^K e^{\alpha_k}$ is the normalizing constant, $K$ denotes the number of candidate operations on one edge and $k$ is the index of the operation.

Given loss $L$, assuming $T = 1$ for simplification, the gradient of $\alpha_k$ is derived as:

$$\begin{aligned}
\frac{\partial L}{\partial \alpha_k} &= \sum_t^T \frac{\partial L}{\partial f^t} \frac{\partial f^t}{\partial \bar{y}^t} \frac{\partial \bar{y}^t}{\partial \alpha_k} \\
&= \frac{\partial L}{\partial f} \frac{\partial f}{\partial \bar{y}} \frac{e^{\alpha_k}}{Z} \left[ y_k - \frac{\sum_k^K (e^{\alpha_k} y_k)}{Z} \right] \\
&= \frac{\partial L}{\partial f} \frac{\partial f}{\partial \bar{y}} \frac{e^{\alpha_k}}{Z} \left( y_k - \bar{y} \right)
\end{aligned} \tag{2}$$

where $f$ is the sign function of the node and for $\frac{\partial f}{\partial \bar{y}}$ we apply a fixed SG function.

For mixed operation at the input of the membrane potential, $\bar{I} = \bar{o}$ and $I = o$. The mixed operation is then:

$$\bar{I} = \sum_k^K \frac{e^{\alpha_k}}{Z} I_k \tag{3}$$

Similarly, the gradient of $\alpha_k$ is then derived as:

$$\begin{aligned}
\frac{\partial L}{\partial \alpha_k} &= \sum_t^T \frac{\partial L}{\partial y^t} \frac{\partial y^t}{\partial u^t} \frac{\partial u^t}{\partial \bar{I}^t} \frac{\partial \bar{I}^t}{\partial \alpha_k} \\
&= \frac{\partial L}{\partial y} \cdot g(u) \cdot 1 \cdot \frac{e^{\alpha_k}}{Z} \left( I_k - \bar{I}_k \right)
\end{aligned} \tag{4}$$

*Equal Contributions. ✉Corresponding author. lengluziwei@huawei.com

36th Conference on Neural Information Processing Systems (NeurIPS 2022).

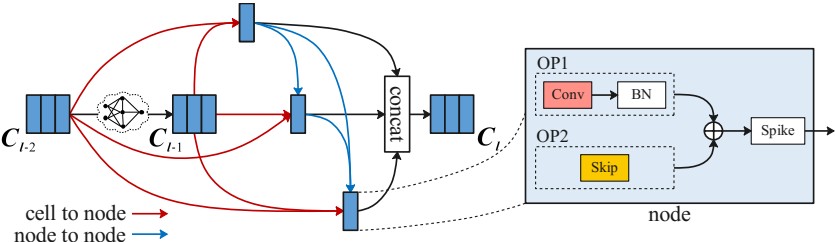

Figure 1: Mixed operation at the input of membrane potential. 2 operations are shown here as an example.

For mixed operation at the spike activation, as Eq. 2 shows, the gradient of $\alpha_k$ stops at $y_k - \bar{y}$, so for candidate operations we can use different SG functions for the corresponding spike activation. However, the learning signal of $\alpha$ is also filtered by an additional SG function $\frac{\partial f}{\partial \bar{y}}$ at the node, which could cause additional noise. In addition, when applied for DGS method, the contribution of $w_{g_i}$ will be filtered by the consecutive spike activation. If the difference between the original weight and $w_{g_i}$ is minor, they could lead to the same spike activation and in the extreme case $y_{g_i} - \bar{y}$ could be 0. For mixed operation at the input of membrane potential, as Eq. 4 shows, the gradient of $\alpha_k$ applies a unified SG function $g(u)$ for different candidate operations which limits the exploration of diverse SG functions. However, since $I_k$ directly depends on the weight, when applied for DGS method it strictly reflects different changes of the original weight, thus giving more accurate learning signals for $\alpha_k$. Also, in forward path this leads to a more concise node without the sign filter, as shown in Fig. 1.

## B  Layer search space

As stated in the main text, to ensure spike-based computation for the network, we apply the nearest interpolation for upsampling in order to maintain a binary feature map. For event-based deep stereo, at the end of each subnetwork, the feature map of the last cell is upsampled to the original resolution. In the feature subnetwork, this is done by upsampling (double scaled) followed by convolution with spike activation, this process is repeated for multiple times until the feature map is recovered to the original resolution. In the matching subnetwork, we cancel the last spike activation and directly output the membrane potential in order to increase the representation ability.

## C  Classification

For classification task, we inherit the searching structure from DARTS [9] and make some adjustments. Same as most NAS structures, we adopt stem layer as our first layer to extract features. Meanwhile it's also a spike emitter to transmit floating numbers to spikes. Then we employ cell structure as our basic unit to construct our whole searching structure. The stem layer takes 3-channel images as input and outputs 48-channel spiking feature maps. For the rest of the network, we employ 6 normal cells and 2 reduction cells. Normal cells keep the dimension of feature maps unchanged, while reduction cells (Cell2 and Cell5) halve the spatial size and double the channel number of former feature maps. Finally, we use a global pooling and a fully-connected layer to end the whole classification process. The auxiliary loss is applied at Cell5. In retraining stage, we use 8 cells as well to avoid some intrinsic problem of DARTS to some extent, like depth-gap between searching structure and retraining structure as described in [4]. In this stage, stem layer uses 108 output channels rather than 48, to extract more features from row image data. The detailed network structure for retraining is shown in Table 1. In extending experiments, we slightly increase the output channel of the first stem layer to 144 and use 3 nodes within a cell, the size of the output feature map of the cell remains the same. Detailed architectures of SpikeDHS-CLA (node=4), SpikeDHS-CLA (node=3) and SpikeDHS-CLA-large (ImageNet) are listed in Table 1, 2 and 3.

Table 1: Network architecture of SpikeDHS-CLA for CIFAR with 4 nodes in a cell.

| Layer | Feature map size $c \times h \times w$ |
|---|---|
| Stem | $108 \times 32 \times 32$ |
| Cell0 | $144 \times 32 \times 32$ |
| Cell1 | $144 \times 32 \times 32$ |
| Cell2 | $288 \times 16 \times 16$ |
| Cell3 | $288 \times 16 \times 16$ |
| Cell4 | $288 \times 16 \times 16$ |
| Cell5 | $576 \times 8 \times 8$ |
| Cell6 | $576 \times 8 \times 8$ |
| Cell7 | $576 \times 8 \times 8$ |
| Pooling | $576 \times 1 \times 1$ |
| FC | 10 |

Table 2: Network architecture of SpikeDHS-CLA for CIFAR with 3 nodes in a cell.

| Layer | Feature map size $c \times h \times w$ |
|---|---|
| Stem | $144 \times 32 \times 32$ |
| Cell0 | $144 \times 32 \times 32$ |
| Cell1 | $144 \times 32 \times 32$ |
| Cell2 | $288 \times 16 \times 16$ |
| Cell3 | $288 \times 16 \times 16$ |
| Cell4 | $288 \times 16 \times 16$ |
| Cell5 | $576 \times 8 \times 8$ |
| Cell6 | $576 \times 8 \times 8$ |
| Cell7 | $576 \times 8 \times 8$ |
| Pooling | $576 \times 1 \times 1$ |
| FC | 10 |

Table 3: Network architecture of SpikeDHS-CLA-large for ImageNet with 3 nodes in a cell.

| Layer | Feature map size $c \times h \times w$ |
|---|---|
| Stem0 | $72 \times 112 \times 112$ |
| Stem1 | $144 \times 56 \times 56$ |
| Cell0 | $144 \times 56 \times 56$ |
| Cell1 | $144 \times 56 \times 56$ |
| Cell2 | $288 \times 28 \times 28$ |
| Cell3 | $288 \times 28 \times 28$ |
| Cell4 | $576 \times 14 \times 14$ |
| Cell5 | $576 \times 14 \times 14$ |
| Cell6 | $576 \times 8 \times 8$ |
| Cell7 | $1152 \times 7 \times 7$ |
| Cell8 | $1152 \times 7 \times 7$ |
| Cell9 | $1152 \times 7 \times 7$ |
| Pooling | $1152 \times 1 \times 1$ |
| FC | 1000 |

# D Event-based deep stereo

We split and preprocess the Indoor Flying dataset from the MVSEC dataset following the same setting as [10, 1, 17]. In split one, 3110 samples from the Indoor Flying 2 and 3 are used as the training set while 861 and 200 samples from the Indoor Flying 1 are used as the test set and validation set. In split three, 2600 samples from the Indoor Flying 2 and 3 are used as the training set while 1343 and 200 samples from the Indoor Flying 1 are used as the test set and validation set.

## D.1 Event encoding

We use stacking based on time (SBT) [11] which merges events into temporally neighboring frames. Specifically, within one stack, a duration of $\Delta t$ event stream is compressed into $n$ frames. The value of each pixel in the $i$th frame is defined as the accumulated polarity of events:

$$P(x, y) = \text{sign}(\sum_{t \in T} p(x, y, t)) \tag{5}$$

where $P$ is the value of the pixel at $(x, y)$, $t$ is the timestamp, $p$ is the polarity of the event and $T \in [\frac{(i-1)\Delta t}{n}, \frac{i\Delta t}{n}]$ is the duration of events merged into one frame. The advantage of SBT is that it is simple and has low computation cost. Besides, event cameras are often embedded with accumulator modules that directly output events in SBT. However, when too few events happening during the interval, SBT may produce a very sparse event image. This limitation can be alleviated for SNN by its intrinsic temporal accumulation effect of the membrane potential. During training, we set $\Delta t = 50\text{ms}$, $n = 5$ and $T = 10\text{ms}$ for each stack and use $6 + 2$ consecutive stacks as one input (with the first 2 stacks as burn in time), corresponding to 6 consecutive ground truth disparities as one label. A training epoch on split one thus contains 518 batches (no time overlap between label batches).

## D.2 Architecture search and retrain

We use random cropping for data augmentation as well as memory saving. The input images are cropped from $260 \times 346$ to $200 \times 280$ with $50\%$ probability at random positions. Our feature subnetwork produces a pair of left and right feature maps $F^L$ and $F^R$. Following volumetric approaches in deep stereo matching [7, 15, 16], we construct a feature volume by concatenating the left feature map and disparity shifted right feature map in channel dimension. With a given disparity shift $d_s \in \{0, 1, 2, \ldots, D\}$, the feature volume for pixel $\mathbf{x}$ can be expressed as:

$$C(\mathbf{x}, d_s) = \text{concat}(F^L(\mathbf{x}), F^R(x - d_s)), C \in \mathbb{R}^{D \times H \times W} \tag{6}$$

For architecture search, we set the channel number of the last stem feature map to 12. In the retraining phase, we expand this channel number from 12 to 24 to improve network performance, the rest of the network also have channel number doubled accordingly. We set $D = 33$, close to the maximum disparity of the MVSEC dataset, which is 37. The detailed network architecture for retraining is shown in Table 4.

## D.3 Random seed experiments

To determine the final architecture, we repeat the experiment for 3 times with different random seeds and select the best architecture based on the validation performance obtained by training from scratch for a short period. Fig. 2 shows that while the architecture exhibits certain sensitivity for initialization, it can be gradually optimized during the search phase. This gradual improvement in dense image prediction may in a degree owing to the layer level optimization. In the classification task, we didn't observe a smooth improvement of the architecture during the search phase. This could be due to the intrinsic optimization problem of the original DARTS [9]. This issue is discussed in [13, 5, 3, 12] where various solutions have proposed. Future works can potentially be improved using these methods.

## D.4 Further ablation study

In the main text, we reported results based on mixed operation at the spike function for the cell (for DGS we applied mixed operation at the membrane potential since there is no multiple spike

Table 4: Detailed architecture of SpikeDHS-Stereo. $C = 12, H = 260, W = 346, D = 33$.

| Module | Layer | Feature map size |
|---|---|---|
| **Feature net** | Stem0 | $C \times H \times W$ |
| | Stem1 | $2C \times \frac{1}{3}H \times \frac{1}{3}W$ |
| | Cell0 | $4C \times \frac{1}{6}H \times \frac{1}{6}W$ |
| | Cell1 | $2C \times \frac{1}{3}H \times \frac{1}{3}W$ |
| | Upsampling | $2C \times \frac{1}{3}H \times \frac{1}{3}W$ |
| **Feature Volume** | Concat | $4C \times D \times \frac{1}{3}H \times \frac{1}{3}W$ |
| **Matching net** | Stem0 | $2C \times D \times \frac{1}{3}H \times \frac{1}{3}W$ |
| | Stem1 | $2C \times D \times \frac{1}{3}H \times \frac{1}{3}W$ |
| | Cell0 | $4C \times \frac{1}{2}D \times \frac{1}{6}H \times \frac{1}{6}W$ |
| | Cell1 | $8C \times \frac{1}{4}D \times \frac{1}{12}H \times \frac{1}{12}W$ |
| | Cell2 | $8C \times \frac{1}{4}D \times \frac{1}{12}H \times \frac{1}{12}W$ |
| | Cell3 | $4C \times \frac{1}{2}D \times \frac{1}{6}H \times \frac{1}{6}W$ |
| | Upsampling | $1 \times D \times \frac{1}{3}H \times \frac{1}{3}W$ |
| **Estimator** | Estimator | $H \times W$ |

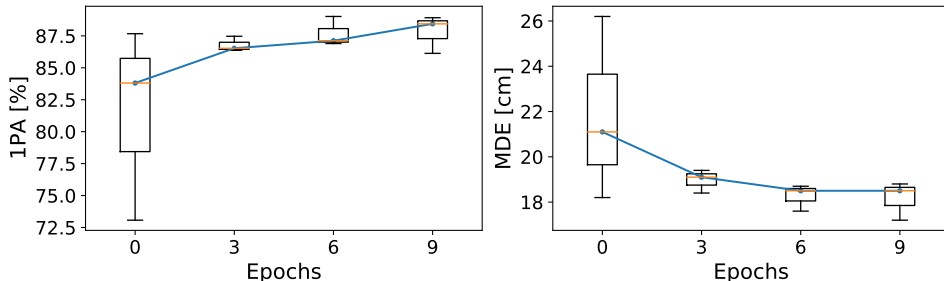

Figure 2: Search progress of SpikeDHS on MVSEC. We record the searched architecture for every 3 epochs. Each recorded architecture is retrained from scratch for 100 epochs and then evaluated on the validation set. We repeat the experiment for 3 times with different random seeds and report the best 1PA and MDE for each recorded architecture.

functions at the stem layer) and results from extending experiments with mixed operation at the membrane potential (MM). To save computation time, we directly convert the searched architecture from mixed at spike to mixed at membrane potential. Specifically, we reinitialize all SG function to $b = 3$ and retrain the model on split one. We also apply DGS in the first stem layer as before. In addition, to compare with ANN converted SNN network, we convert the DDES model to spike-based model, i.e replace Relu units with LIF neurons, to create DDES-SNN model and directly train it. The results are summarized in Table 5. Models with mixed operation at the membrane potential achieves significantly better results, probably due to a more concise node which uses fewer SG function. The performance of the model is further improved with DGS method, demonstrating the effectiveness of the algorithm. The DDES-SNN model performs much worse than SpikeDHS-Stereo, though additional fine tuning could potentially improve its performance, it shows the difficulty of directly applying ANN architecture for spiking neurons in dense prediction task.

### D.5 FPS calculation

For DARTS-SNN, We use $8$ consecutive stacks as one input and calculate corresponding inference time of the network. Then we divide the time by $8$ to obtain the inference time of producing one disparity map. For DDES we run the public available code with one input sample and produce one disparity map. Both experiments are repeated multiple times to obtain the average duration. These experiments are performed on a single NVIDIA Tesla V100 (32G) GPU.

Table 5: We denote the best and second best results in bold and underscore. Symbols meaning: (·), streaming tests;$^D$, training with DGS; MM, mixed operation at the membrane potential

| Method | EO | No. param. | MDE [cm] ↓ | | Median depth error [cm] ↓ | | Mean disparity error [pix] ↓ | | 1PA [%] ↑ | |
| --- | --- | --- | --- | --- | --- | --- | --- | --- | --- | --- |
| | | | split1 | split3 | split1 | split3 | split1 | split3 | split1 | split3 |
| DDES [10] | ✓ | 2.33M | 16.7 | 27.8 | 6.8 | 14.7 | 0.59 | 0.94 | 89.4 | 74.8 |
| SpikeDHS | ✓ | 0.87M | 16.5(16.5) | 19.4(19.5) | 6.5(6.5) | 10.6(10.6) | 0.57(0.57) | 0.73(0.74) | 90.1(90.2) | 88.5(88.4) |
| SpikeDHS$^D$ | ✓ | 0.87M | 15.9(15.8) | 19.1(19.3) | 6.3(6.3) | 10.4(10.5) | 0.54(0.54) | 0.72(0.74) | 90.7(90.8) | 88.9(88.8) |
| DDES-SNN | ✓ | 2.33M | 53.4 | - | 37.8 | - | 2.08 | - | 37.1 | - |
| SpikeDHS (MM) | ✓ | 0.87M | 15.7(15.7) | - | 6.3(6.3) | - | 0.55(0.54) | - | 91.0(91.1) | - |
| SpikeDHS$^D$ (MM) | ✓ | 0.87M | 15.4(15.4) | - | 6.0(6.0) | - | 0.54(0.54) | - | 91.3(91.4) | - |

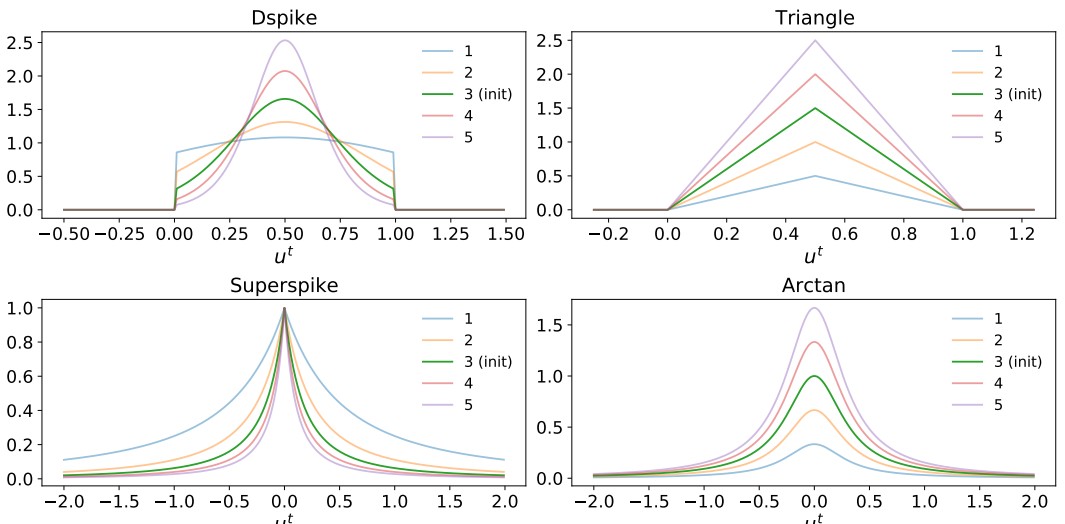

Figure 3: The shape of different SG functions with different temperature factor $b$. We set the initial value of $b$ to 3.

### D.6 Different SGs for DGS

We evaluate four different SG functions including Dspike [8] (Eq. 7), Triangle [2] (Eq. 8), Superspike [14] (Eq. 9) and Arctan [6] (Eq. 10) functions with fixed hyperparameters during training and varying hyperparameters with DGS. These functions are described as following

$$\delta'_D(x) = \frac{b}{2} \cdot \frac{1 - \tanh^2(b(x - \frac{1}{2}))}{\tanh(\frac{b}{2})} \text{ if } 0 \leq x \leq 1 \tag{7}$$

$$\delta'_T(x) = b \cdot \max\{0, 1 - |x - \frac{1}{2}|\} \tag{8}$$

$$\delta'_S(x) = \frac{1}{(b \cdot |x| + 1)^2} \tag{9}$$

$$\delta'_A(x) = \frac{b/3}{1 + (\pi x)^2} \tag{10}$$

Each SG function has a temperature factor $b$ to control its shape through DGS. A visualization is shown in Fig. 3.