# OpenReview forum: "Differentiable hierarchical and surrogate gradient search for spiking neural networks"
_NeurIPS.cc/2022/Conference — NeurIPS 2022 Accept_

### Official Review · Reviewer_LhUf · 2022-06-30

**Rating:** 8
**Confidence:** 5
**Soundness:** 3 good
**Presentation:** 3 good
**Contribution:** 4 excellent

**Summary:**

In this submission draft, the authors device a differentiable hierarchical search framework tailored for SNNs. In the meantime, this framework is able to search the surrogate gradient in a differentiable manner. Their methods are validated on the CIFAR dataset and an event-based deep stereo dataset.

**Questions:**

1. Can authors elaborate on the search space in this work, especially when compared to the search space in DARTS and NAS-Bench-101?

2. Can the authors provide more details on the difference in searching gradients with Dspike in Section 3.4? Currently, the only saying is *" [31] demonstrates that by optimizing the width (or temperature) of the SG function the performance of SNN can be improved"*, but there is no drawbacks analysis and to which degree that this DGS method can improve.

**Limitations:**

1. Need to include two prior SNN NAS papers in the discussion or experiments. See references below.


2. A critical problem is that there is no comparison between the searched architecture and the ResNets used in other works. What if the searched architecture has a higher capacity than ResNets?

3. An ablation study on the DGS is recommended, the authors should compare static temperature gradient, DGS, and [31] on the same neural architecture and under the same training receipt.

4. Better to have an ImageNet result.



------

**References**

Na B, Mok J, Park S, et al. AutoSNN: Towards Energy-Efficient Spiking Neural Networks[J]. arXiv preprint arXiv:2201.12738, 2022.

Kim Y, Li Y, Park H, et al. Neural architecture search for spiking neural networks[J]. arXiv preprint arXiv:2201.10355, 2022.

**Strengths And Weaknesses:**

Overall this is an interesting work. The authors come up with an end-to-end differentiable framework that solves two critical problems in SNN: the architecture and the surrogate gradient.


1. Developing SNN-oriented architectures are novel and necessary. Even though this work is not the first trial in the community.
2. Searching the SG is interesting and I am glad to see a learning-based method to address the issue.
2. The results on the CIFAR10/100 dataset are promising.

---

> ### Author Response · Authors · 2022-08-02
> **Reply to Reviewer LhUf**
>
> Thank you for your recognition of the contribution of our work and detailed review. Here we provide replies to your concerns and really appreciate it if you can kindly reconsider your ratings. If you have further questions, please do not hesitate to reply to us.
>
>
> **For your questions**:
>
>
> **Q1**: Due to the large training time cost of deep SNNs, we limit to a small search space. For CIFAR, we use the same downsampling network backbone as DARTS [1] and limited our candidate operation to {skip, 3x3 conv with Dspike(b=3), 3x3 conv with Dspike(b=5)}. Comparatively, DARTS [1] used different types of convolutions and pooling operations, including {3 × 3 and 5 × 5 separable conv, 3 × 3 and 5 × 5 dilated separable conv, 3 × 3 max pooling, 3 × 3 average pooling, identity}. For event-stereo, our search space includes both the cell operations and the layer structure, where the latter is selected among {upsampling, downsampling, same}. Similar to DARTS and our work, NAS-Bench-101 used a stem followed by stacks of directed acyclic graph cells as network pattern and searched for optimal cell structures. However, it constructed a map from exhaustive CNN architectures (423k structures) to their runtime and accuracies on CIFAR10, which enabled NAS experiments to be run via querying a table instead of performing the costly train and evaluate procedure. Similar works can be done for SNNs in the future.
>
>
> **Q2**: This is a good question. Empirically, we found that search grid intervals ($\Delta b$) and applied epoch interval ($e_D$) will influence the effect of DGS. The selection of different SG functions and applied layers also have influence (please see replied tables for **Q1** Reviewer hUzC and general response). In principle, the degree of improvement of DGS is largely related to how much the performance of SNN is influenced by an ill SG function. We will add more analysis in future version.
>
>
> **For limitations**:
>
>
> **L1**.  Thank you for reminding us the two contemporary works about NAS for SNN, for the comparison between theirs and our work, please see our general response. We will add the comparison in revision.
>
>
> **L2**: A comparison of network capacity is shown in table below. We calculated the capacity of ResNet-19 from the open source code of TET. For Dspike, we didn't find the open source code so we don’t know the exact capacity of the ResNet-18 used in the paper. However, they used a modified version from the Resnet-18 in [2] and we constructed the network following exactly their descriptions. In general our network has similar capacities as ResNets we compared with.
>
>
> | Architecture | Method | Simulation length | Params.[M] | Accuracy [%] |
> |  :----: | :----: | :----: | :----: | :----: |
> | ResNet-18 | FDG | 6 | 11.45 | $94.25 \pm 0.07$ |
> | | static | 6 | 11.45 | $94.54$ |
> | | DGS | 6 | 11.45 | $94.66$ |
> | ResNet-19 | TET | 6 | 12.63 | $94.50 \pm 0.07$ |
> | DARTS-SNN | DARTS-SNN   |  6 | 12.33 | $94.34\pm 0.06$ |
> |  | DARTS-SNN^D  | 6 | 12.33 | $94.68\pm 0.05$ |
>
>
> **L3**: A table of comparison of FDG (Dspike with varying temperature), static training (Dspike with fixed temperature) and DGS based on ResNet-18 (see **L2** for how we construct the network) is shown in table above. Since we didn't find open source code of Dspike paper and we were not able to obtain ideal results with FDG implemented by ourselves, using exactly the same training receipt is challenging. The result of FDG is taken from the original paper. For DARTS-SNN we didn't use techniques such as initialization model or time-inheritance training as in FDG. Another difference is that FDG uses tdBN and we use normal batch normalization during training, whose parameters are converted to convolution weights during inference.
>
>
> **L4**: With limited time our experiments on ImageNet is preliminary, initial results are shown in table below. By the time of submission the experiment of DGS is still running so we provide its latest result. We assume for such a big dataset more explorations on hyperparameters and candidate operations are necessary to obtain competitive results with current SOTA SNNs using advanced training techniques. We believe this should not undermine the contribution of this work.
>
>
> | Architecture | Method | Simulation length  | Trained epochs | Top-1 accuracy [%] |
> |  :----: | :----: | :----: | :----: | :----: |
> | DARTS-SNN | DARTS-SNN   | 6 | 100 | 67.96 |
> |  |    |  6 |  55 | 64.63 |
> |  DARTS-SNN | DARTS-SNN^D  | 6 | 55 | 65.19 |
>
>
> [1] Hanxiao Liu, Karen Simonyan, and Yiming Yang. Darts: Differentiable architecture search. arXiv preprint arXiv:1806.09055, 2018.
>
>
> [2] Tong He, Zhi Zhang, Hang Zhang, Zhongyue Zhang, Junyuan Xie, and Mu Li. Bag of tricks for image classification with convolutional neural networks. In Proceedings of the IEEE/CVF Conference on Computer Vision and Pattern Recognition, pages 558–567, 2019.

---

> > ### Comment · Reviewer_LhUf · 2022-08-03
> > **Haven‘t found the general response yet.**
> >
> > I'd like to thank the authors for their effortful, detailed response. However, there is no general response yet. Can the authors provide it so that I can re-evaluate the score?

---

> > > ### Author Response · Authors · 2022-08-03
> > > **General response added**
> > >
> > > Thank you for your remind and quick response, we have updated the general response (turns out we didn't make it visible to all Reviewers in the last version). Please contact us for potential further questions.

---

> > > > ### Comment · Reviewer_LhUf · 2022-08-03
> > > > **Post-rebuttal Review**
> > > >
> > > > The authors have honestly addressed my concerns, I raise to score to 8 and hope to see this paper in the NeurIPS 2022 conference.
> > > >
> > > > In their future revision, I hope they can add ImageNet results, comparison to literature and architecture details. If possible, the energy estimation of the searched architectures should also be added into comparison.

---

> > > > > ### Author Response · Authors · 2022-08-04
> > > > > **Response to Reviewer LhUf**
> > > > >
> > > > > Thank you for your reevaluation and recognition of our work. We will add ImageNet results, corresponding comparisons and energy estimation in our future revision. Thanks again.

---

### Official Review · Reviewer_hUzC · 2022-07-01

**Rating:** 6
**Confidence:** 3
**Soundness:** 3 good
**Presentation:** 2 fair
**Contribution:** 2 fair

**Summary:**

In this work, the authors propose a differentiable hierarchical search framework for spiking neurons, where spike-based computation is realized on both the cell and the layer level search space. Meanwhile, the authors find effective SNN architectures under limited computation cost. In order to avoid the standard SG approach that leads the network into suboptimal solutions, the authors propose a differentiable surrogate gradient search method where the SG function can be efficiently optimized locally in parallel. Finally, this work shows some interesting results on the image classification tasks.

**Questions:**

1. My core concern lies in the choice of the surrogate gradient (SG) function in the methods section. First, it looks like the SG function is from existing work [31]. If this is true, the reviewer suggests that the authors put this subsection in the background section. Second, what is the motivation for choosing such a SG function? Because, the motivation for the design of the SG function has been given in the existing studies [1-2]. It is clear that SG methods are a key part of the direct training of spiking neurons, and the choice of such methods also affects the search results, and the reviewers suggest that the authors explain the motivation clearly.

[1]"Accurate and efficient time-domain classification with adaptive spiking recurrent neural networks," Nat. Mach. Intell. 3(10): 905-913 (2021)

[2]"A Hybrid Spiking Neurons Embedded LSTM Network for Multivariate Time Series Learning under Concept-drift Environment," in IEEE Transactions on Knowledge and Data Engineering, doi: 10.1109/TKDE.2022.3178176.

2. The parallel optimization involved in the search process needs further explanation by the authors.

3. There is a significant difference in performance improvement on the CIFAR-10 and CIFAR-100 datasets, and the authors need to explain the reason for this phenomenon.

4. The ablation experiments concluded the important value of temporal dynamics in the spiking neuron, but why the MDE metric and Median depth error metric appear opposite on split3. i.e., DARTS-BN^D better than DARTS-SNN^D in terms of Median depth error metric-split3.

5. In Table III, the experimental results show that the power consumption of the proposed method is significantly lower. However, the reviewers note that the SNN model is tested on a chip and whether it is the device that leads to a more recent reduction in power consumption.


**Limitations:**

The authors illustrate the limitations of their work.

**Strengths And Weaknesses:**

Strengths:
1. A hierarchical differentiable surrogate gradient search framework is proposed to obtain better performance of the spiking model.
2. Significant improvements in energy savings on deep stereo.

Weakness：
1. In terms of writing, some methods that were not proposed in the work were placed in the methods section. There are also some typos in terminology.
2. The results of the ablation experiments and the analysis of some elements do not match.
3. The font of the figure seems to be a small and not clear enough, which leads to a very careful reading to find valuable information.
4. The percentage improvement of the proposed method varies greatly on the two image classification datasets. Even the improvement on CIFAR-10 is only 0.18.

---

> ### Author Response · Authors · 2022-08-02
> **Reply to Reviewer hUzC**
>
> Thank you for your careful review and nice suggestions, we will recheck typos and terminologies and enlarge fontsize in the figure in revision. Here we provide exact replies to your concerns and really appreciate it if you can kindly reconsider your ratings. If you have further questions, please do not hesitate to reply to us.
>
>
> **For your questions:**
>
>
> **Q1**: It is a good suggestion to put the SG function in the background section. We will put sec 3.1 as preliminary section. We will also add [1] and [2] as references for SG function in the background. Our initial motivation of using the Dspike function is because it can cover a wide range of SGs with different smoothness by changing the temperature. In principle other SG functions with similar property could also be considered. However, as analyzed in [1], it is possible that SG functions with penalization on extreme values can lead to better results. The table below shows our preliminary results with different SG functions for event-based stereo task on split 1. Dspike and triangle functions (both with zero values for region outside of [0,1]) outperforms the other two functions where both have long tails expanding to extreme values. Note that we apply mixed operation at the membrane potential (see supplement D.4) for this task. We will add more analysis in revision.
>
>
> | SG function |   Method   |  1PA [%]  ↑|
> | :----:     |   :----:   |       :----: |
> | Dspike |   fixed hyperparameter   | 91.0  |
> |  |   DGS    |  91.3  |
> | triangle |   fixed hyperparameter   | 90.9  |
> |  |   DGS    |  91.3  |
> | Arctan |   fixed hyperparameter   | 89.3   |
> |  |   DGS    |  90.1  |
> | Superspike |   fixed hyperparameter   | 89.6   |
> |  |   DGS    |  89.7  |
>
>
> **Q2**: For parallel optimization of SG functions in DGS method, we denote a special case when the chosen SG function is linearly scaled with its control parameter (Line 190), e.g rectangle functions with the same width but different amplitudes, thus the weights updated with different SGs can be estimated using one SG value, enabling fast parallel update of SG functions across different layers. However, in experiment we only applied DGS in one layer. We will clarify it in revision.
>
>
> **Q3**: For both CIFAR10 and CIFAR100, we compare with the most recent SOTA SNNs with advanced training strategies that we didn’t use, such as the moment loss in TET and initialization model as well as time-inheritance training in Dpike. It would not be unexpected that our method had different improvements on two different datasets.
>
>
> **Q4**: There are four metrics in the evaluation criteria of the MVSEC dataset and the overall performance of SNN is better than BNN on both splits, either with DGS or not. For the issue on median depth error, the loss function (Eq. 7) we used is designed to reduce the disparity error, i.e pixel error of the disparity map, and it is usually the case when 1PE gets its minimum other metrics are not, since they are calculated with different methods.
>
>
> **Q5**: The measure of power consumption is performed based on unified method for both ANN and SNN, i.e. by counting the MAC numbers of the network. SNN has significantly lower energy cost since the network has sparse activation and only addition operation, which consumes less energy compared to multiplication operation (0.9 pJ vs 4.6 pJ in 45nm CMOS technology [1, 2]). However, future implementation of the SNN on potential neuromorphic device could lead to further reduction of energy cost.
>
>
> [1] M. Horowitz, “1.1 Computing’s energy problem (and what we can do about it),” in IEEE Int. Solid-State Circuits Conf. (ISSCC) Dig. Tech. Papers, Feb. 2014, pp. 10–14.
> [2] Nitin Rathi and Kaushik Roy. Diet-snn: A low-latency spiking neural network with direct input encoding and leakage and threshold optimization. IEEE Transactions on Neural Networks and Learning Systems, 2021.451

---

> > ### Comment · Reviewer_hUzC · 2022-08-05
> > **New Comments**
> >
> > Thank you very much for the author's reply and efforts to supplement the experiment.
> >
> > According to the answer of Q5, then the exploration of SNN in neuromorphic hardware means that it has not been carried out. Computational efficiency is only analyzed from the perspective of addition and multiplication operations.
> >
> > Thus, I maintain my rating based on the responses.

---

> > > ### Author Response · Authors · 2022-08-05
> > > **Reply to new comments from Reviewer hUzC**
> > >
> > > Thank you for your reply and recognition of our efforts on extended experiments. We respect your reevaluation, however we really hope that you can reconsider your ratings, our reasons are following:
> > >
> > >
> > > 1. For the energy estimation, we followed the approach [1] as applied in Diet-SNN (IEEE TNNLS 2021) [2] and Dspike (NeuraIPS 2021) [3], which is a commonly recognized way for power consumption estimation of SNN. We acknowledge the significance of a power measurement on real neuromorphic hardware however this would be another work (considering the versatility of neuromorphic chips and their constrains on SNN implementation) and is not the main contribution of this work. If the reviewer prefer, we can consider adding estimated energy cost on neurormorphic hardware based their energy cost per spike from published literatures.
> > >
> > >
> > > 2. On algorithm level, as strengthened in the general response, using the same GPU (NVIDIA Tesla V100) our approach surpassed sophisticated ANN in benchmark event-based stereo task in terms of accuracy and inference speed, with much smaller network size. We believe this is already an achievement and a breakthrough in the field of SNN, since only a few SNN works are applied to hard dense prediction problems and the current SOTA are significantly behind ANN performance ([4, 5, 6]).
> > >
> > >
> > > [1] M. Horowitz, “1.1 Computing’s energy problem (and what we can do about it),” in IEEE Int. Solid-State Circuits Conf. (ISSCC) Dig. Tech. Papers, Feb. 2014, pp. 10–14.
> > >
> > >
> > > [2] Nitin Rathi and Kaushik Roy. Diet-snn: A low-latency spiking neural network with direct input encoding and leakage and threshold optimization. IEEE Transactions on Neural Networks and Learning Systems, 2021.451
> > >
> > >
> > > [3] Yuhang Li, Yufei Guo, Shanghang Zhang, Shikuang Deng, Yongqing Hai, and Shi Gu. Differentiable spike: Rethinking gradient-descent for training spiking neural networks. Advances in Neural Information Processing Systems, 34, 2021.
> > >
> > >
> > > [4] Jesse Hagenaars, Federico Paredes-Vallés, and Guido De Croon. Self-supervised learning of event-based optical ﬂow with spiking neural networks. NeuraIPS 2021.
> > >
> > >
> > > [5] Youngeun Kim, Joshua Chough, and Priyadarshini Panda. Beyond classifcation: Directly training spiking neural networks for semantic segmentation. arXiv preprint arXiv:2110.07742, 2021.
> > >
> > >
> > > [6] Ulysse Rançon, Javier Cuadrado-Anibarro, Benoit R Cottereau, and Timothée Masquelier. Stereospike: Depth learning with a spiking neural network. arXiv preprint arXiv:2109.13751, 2021.

---

### Official Review · Reviewer_cjsQ · 2022-07-08

**Rating:** 6
**Confidence:** 3
**Soundness:** 3 good
**Presentation:** 3 good
**Contribution:** 3 good

**Summary:**

This work is aimed to search for both the optimal SNN architecture and hyperparameters of surrogate gradient (SG) functions. In the architecture search phase, they use DARTS and refine the search to different granularities (layer-level and cell-level). The search for SG function (DGS) focuses on optimizing the temperature of the Dspike SG function. The results show that searched architecture achieve SOTA  performance on image classification and event-based stereo matching task.

**Questions:**

1. The comparison of previous NAS work on SNNs should be added.
2. The training pipeline is unclear to me. Does the training pipeline include **both** ANN-to-SNN conversion and direct training using SG? I notice that L158 suggests that ANN layers are converted to SNN layers. If so, the comparison to some previous work using only directly trained SNNs is not fair.
3. Why Dspike is the only tested SG function? As mentioned in ref. 64, the choice of SG function affects the viable range of hyperparameters. The range determines the difficulty level of search. As there has been no claimed best SG function till now and ref. 64 points out that some different SG functions actually have the same maximum performance, the author should try some popular SG functions like SuperSpike[1], triangle[2], ArcTan[3], etc., to show the versatility of this method.

[1] Friedemann Zenke, and Surya Ganguli. "Superspike: Supervised learning in multilayer spiking neural networks." *Neural computation* 30.6 (2018): 1514-1541.

[2] Guillaume Bellec, et al. "Long short-term memory and learning-to-learn in networks of spiking neurons." *NeurIPS* (2018).

[3] Wei Fang, et al. "Incorporating learnable membrane time constant to enhance learning of spiking neural networks." *ICCV*. 2021.

**Strengths And Weaknesses:**

**Pros**
1. The search for the architecture alone significantly increases the performance of image classification tasks, which reveals the potential to be applied to various more complicated tasks.
2. The idea of searching hyperparameter of SG function is novel, simple but effective.

**Cons**
1. The idea of applying NAS on SNNs is not novel till the deadline of NeurIPS submission. SNASNet[1] and AutoSNN[2] have proposed that NAS methods can be used for searching the structure of SNNs. The latter has been accepted at ICML2022.
2. The articulation of the training pipeline is not highlighted and is somewhat unclear to me. See the **Questions** below.
3. The trials of search on SG functions are confined to the Dspike function.

[1] Youngeun Kim, et al. "Neural architecture search for spiking neural networks." *arXiv preprint arXiv:2201.10355* (2022).

[2] Byunggook Na, et al. "AutoSNN: Towards Energy-Efficient Spiking Neural Networks." *arXiv preprint arXiv:2201.12738* (2022).

---

> ### Author Response · Authors · 2022-08-02
> **Reply to Reviewer cjsQ**
>
> Thank you for your thorough review on our work and constructive questions. Here we provide exact replies to your questions and really appreciate it if you can kindly reconsider your ratings. If you have further questions, please do not hesitate to reply to us.
>
>
> **For your questions:**
>
>
> **Q1**: Thank you for reminding us the two contemporary works about NAS for SNN, for the comparison between theirs and our work, please see our general response. We will add the comparison in revision.
>
>
> **Q2**:  During the search phase, we use Relu activation function for stem layers, which constitute only a small part of the model parameters (0.03% for classification and 5% for event-stereo), the majority of the network (cells) is SNN. In retraining phase we replace the Relu function with spiking function and retrain the searched model from scratch without inheriting any weights from the searched architecture. Since we directly train a full SNN, we think its fair enough to compare with directly trained SNNs. The reason for using Relu activation is to ensure more stable updating of the supernet, since deep SNNs suffer from gradient vanish problem when the SG functions is not chosen appropriately. Empirically we replaced the Relu function in stem layer with Dspike function under appropriate hyperparameters and the search process is also stable. We will add it in revision.
>
>
> **Q3**: This is a very good suggestion. We chose Dspike as SG function because it can cover a broad range of SGs with different smoothness by changing the temperature. However, other SG functions with similar property can also be considered. In preliminary experiments we tested with Superspike, triangle and Arctan SG functions with fixed hyperparameter training and varying hyperparameters with DGS. The table below shows results with different SG functions for event-based stereo task on split 1, which demonstrates the robustness of DGS for different SG functions. We will add more details in revision.
>
>
> | SG function |   Method   |  1PA [%]  ↑|
> | :----:     |   :----:   |       :----: |
> | triangle |   fixed hyperparameter   | 90.9  |
> |  |   DGS    |  91.3  |
> | Arctan |   fixed hyperparameter   | 89.3   |
> |  |   DGS    |  90.1  |
> | Superspike |   fixed hyperparameter   | 89.6   |
> |  |   DGS    |  89.7  |

---

> > ### Comment · Reviewer_cjsQ · 2022-08-05
> > **About the stem layers in SNNs**
> >
> > Thanks for the authors' clarification and effort in supplement experiments.
> >
> > However, I still have questions about how the choice of stem layers is decided and why they are treated separately from the other SNN layers.

---

> > > ### Author Response · Authors · 2022-08-05
> > > **Reply about the stem layers in SNNs**
> > >
> > > Thank you for your reply and thoughtful question. Stem layers are common in NAS methods and we inherited this structure. They are normal convolution layers for channel variation and feature extraction. For classification task, we didn't choose it specifically and used one stem layer as the first layer, like the original DARTS [1] framework. It maps the input image with dimension [3, H, W] to a feature map of size [108, H, W]. For event-stereo task two stem layers were used as front layers for each subnetwork and the choice is empirical. Other reasonable choices of number of stem layers can be explored however it would be less related to the core contribution of this work.
> > >
> > >
> > > In terms of why stem layers are treated separately from other SNN layers, as we have stated before, the stem layers were applied with Relu activation during search and in retraining they were switched to spiking activation for full SNN training. The reason why we chose Relu rather than spiking activation for stem is because different from cell layers where SG functions can be optimized for spiking activation, here the SG function is fixed (only during search, in retraining DGS can be applied), so if the SG is not chosen appropriately it can result in unstable training or gradient vanish for the supernet. However when the SG function is appropriately chosen (may need trial and error since it could vary for different tasks and network structures) this phenomenon can also be avoided. This has been demonstrated in experiments and we can add it in revision.
> > >
> > >
> > > In brief summary, the use of Relu rather than spiking activation for stem is to **ensure an efficient search of SNN cell structure, avoiding potential influence from an ill chosen SG function of the stem**. However this does not mean we have to use Relu, i.e separating stem from other SNN layers, appropriate SG functions for the stem can be obtained with sufficient empirical exploration. We hope this answers your question, please contact us if you have further questions.
> > >
> > >
> > > [1] Hanxiao Liu, Karen Simonyan, and Yiming Yang. Darts: Differentiable architecture search. arXiv preprint arXiv:1806.09055, 2018.

---

> > > > ### Comment · Reviewer_cjsQ · 2022-08-08
> > > > **Reevaluation of score**
> > > >
> > > > The authors have well addressed most of my concerns. The newly added comparison in general response shows DGS has outperformed previous studies. The supplement experiments on a variety of SG functions substantiate the versatility and make their work more solid.
> > > >
> > > > For the above reason, I will raise my score to 6. I suggest these supplement trials be illustrated in the main text in their future revisions.

---

> > > > > ### Author Response · Authors · 2022-08-08
> > > > > **Reply to reevaluation of Reviewer cjsQ**
> > > > >
> > > > > Thank you for your reevaluation and recognition of our work. We will illustrate these supplement experiments in the main text of our future revision. Thanks again.

---

### Official Review · Reviewer_Y1Po · 2022-07-12

**Rating:** 5
**Confidence:** 4
**Soundness:** 2 fair
**Presentation:** 2 fair
**Contribution:** 2 fair

**Summary:**

The paper presents a NAS optimization algorithm for SNN search.

**Questions:**

See my above comments.

**Limitations:**

See weakness section.

**Strengths And Weaknesses:**

+The authors present interesting results with the differentiable NAS search.
-There are two major works related to NAS for SNNs that has been recently out [5], [11]. The authors have not cited these works. It makes me wonder what is the author's contributiona s compared to these works. [5]  talks about the fact that training SNN using standard NAS methods might be too complex because SNNs need large training time, so they come up with a NAS without tarining technique. [11] talks about a differentiable NAS technique. Both works show good results on a avraiety of datasets, and talk about the intricacies of architecture search.
-The authors have also compared their technique to select works in table 1.  There is a lot of work from Priya Panda's group at Yale, Emre Neftci's group, and many others with regard to SNN training that show SOTA results on DVS and static datasets. The authors have failed to acknowledge most recent works.

Below is a list of publications (not exhaustive) that the author should check:
[1] Towards spike-based machine intelligence with neuromorphic computing K Roy, A Jaiswal, P Panda Nature 575 (7784), 607-617

[2] Enabling spike-based backpropagation for training deep neural network architectures C Lee, SS Sarwar, P Panda, G Srinivasan, K Roy Frontiers in neuroscience, 119

[3] Rate Coding Or Direct Coding: Which One Is Better For Accurate, Robust, And Energy-Efficient Spiking Neural Networks? Y Kim, H Park, A Moitra, A Bhattacharjee, Y Venkatesha, P Panda ICASSP 2022-2022

[4] Neuromorphic Data Augmentation for Training Spiking Neural Networks Y Li, Y Kim, H Park, T Geller, P Panda arXiv preprint arXiv:2203.06145

[5] Neural architecture search for spiking neural networks Y Kim, Y Li, H Park, Y Venkatesha, P Panda arXiv preprint arXiv:2201.10355

[6] Optimizing deeper spiking neural networks for dynamic vision sensing Y Kim, P Panda Neural Networks 144, 686-698

[7] Federated Learning with Spiking Neural Networks Y Venkatesha, Y Kim, L Tassiulas, P Pand IEEE Transactions on Signal Processing 2021

[8] Beyond classification: directly training spiking neural networks for semantic segmentation Y Kim, J Chough, P Panda arXiv preprint arXiv:2110.07742

[9] Revisiting batch normalization for training low-latency deep spiking neural networks from scratch Y Kim, P Panda Frontiers in neuroscience, 1638

[10]Na, Byunggook, et al. "AutoSNN: Towards Energy-Efficient Spiking Neural Networks." arXiv preprint arXiv:2201.12738 (2022).

---

> ### Author Response · Authors · 2022-08-02
> **Reply to Reviewer Y1Po**
>
> Thank you for your review on our work and the list of recommended papers. Here we provide exact replies to your concerns and really appreciate it if you can kindly reconsider your ratings. If you have further questions, please do not hesitate to reply to us.
>
>
> Thank you for reminding us the two contemporary works about NAS for SNN. Regarding differences and the contribution of our work, please see our general response. Related contents will be added in revision.
>
>
> In terms of "The authors have compared their technique… The authors have failed to acknowledge most recent works.":
> within limited space, we only presented the most recent publications of SNN classification on CIFAR dataset from top conferences (Dspike, 2021 NeuraIPS; TET, 2022 ICLR, etc) and they are the current SOTA in terms of accuracy (as far as we know) by submission. We acknowledge works from Priya Panda's group and Emre Neftci's group (actually we have already cited two of their works, Line 75 as [41] and [42]) and will cite more of their works in revision to broaden the scope of the paper.
>
>
> Thank you for providing the check list of papers and we have checked each of them. Except the two NAS SNN papers ([5] and [11]) mentioned above, [8] was already cited in our manuscript (Line 96 as [23]). [1] is a nice perspective paper on neuromorphic computing and we can cite it. [2] is about SNN training with SG method and [6] developed several techniques for SNN on event-based image classification, we could consider cite them as well. However, the rest papers in the list are comparatively less related to our work. We acknowledge previous works in various aspects of SNNs from Priya Panda's group and in principle we would like to cite as much related papers as possible to broaden the scope of our paper.

---

> > ### Comment · Reviewer_Y1Po · 2022-08-08
> > **About the comparison between DARTS-SNN and other methods**
> >
> > Dear authors,
> >
> > Thank you for doing a revised analysis to showcase the accuracy performance improvement of your work over previous NAS studies on SNNs. You have shown number of spikes as a comparison for energy efficiency. However, analytical estimations using the methodology of [1,2] on 45nm CMOS is a gross estimation  and really does not qualify as efficiency improvement. For e.g. you might be sparse, but if your hardware cannot take advantage of sparsity, you will in fact have more computations. I think more than inference time sparsity, one major bottleneck of using NAS which previous works like SNASNet and AutoSNN have acknowledge is the search time complexity. This is a huge bottleneck, and just throwing compute for doing DARTS may not be the most reasonable for SNNs given that there will be exploding memory usage with SNNs irrespective of sparsity if you are running on GPU platform on Tensorflow/Pytorch. Thus, SNN based NAS will be more expensive in terms of training complexity than ANN NAS. So, I am not sure if inference time sparsity is a good and strong argument to make as a major contribution for the paper. The authors make an argument that their method does well on event-stereo tasks compared to ANNs. Again, there have been some recent work on neuromorphic datasets that show SOTA performance with SNNs.
> >
> > I still think that with respect to previous works as the authors is marginal. The authors just use a previous NAS technique and show accuracy improvement without actually looking at the bottomline problems of search time cost, SNN computation overload during search etc. The inference time sparsity by just counting number of spikes is a very gross approximation taht cannot be used to justify the novelty of this work. And hence I still recommend for rejection.
> >
> > [1] M. Horowitz, “1.1 Computing’s energy problem (and what we can do about it),” in IEEE Int. Solid-State Circuits Conf. (ISSCC) Dig. Tech. Papers, Feb. 2014, pp. 10–14.
> >
> > [2] Nitin Rathi and Kaushik Roy. Diet-snn: A low-latency spiking neural network with direct input encoding and leakage and threshold optimization. IEEE Transactions on Neural Networks and Learning Systems, 2021.451

---

> > > ### Author Response · Authors · 2022-08-09
> > > **Reply to new comment from Reviewer Y1Po**
> > >
> > > Thank you for your reply. We understand your concerns for search time cost and power consumption measurement problems in the field of SNN. However, we believe these issues should not influence the main contribution of this work stated in the general response, the reasons are following:
> > >
> > >
> > > **For the issue of search time cost**
> > >
> > >
> > > In terms of "The authors just use a previous NAS technique and show accuracy improvement without actually looking at the bottomline problems of search time cost, SNN computation overload during search etc": Actually, our work has considered the search time cost problem. One of our motivation of using continuous differentiable architecture search (DARTS) methods rather than traditional discrete
> > > NAS methods is to reduce search time cost (explained in line 66-71 of the manuscript). Experimentally, by reducing the number of nodes in cell and using limited candidate operations, our approach **demonstrates limited computation cost** in both classification (1.4 GPU day, line 210-211, our new experiments with 3 nodes (n=3 experiments in general response) cost 0.5 day) and event-stereo task (0.4 GPU day, line 246-247) **meanwhile achieving high accuracy**.
> > >
> > >
> > > In terms of "Thus, SNN based NAS will be more expensive in terms of training complexity than ANN NAS.": A main reason for the larger training time cost of SNN compared to ANN is that SNN needs multiple time steps (T) for evolution. In event-based stereo, **we avoid this problem by using streaming training with T=1** (line 234-236), thus largely reducing the searching and retraining time cost of SNN, demonstrating the efficiency and low latency of SNN in processing temporal data streams.
> > >
> > >
> > > **For the issue of power consumption**
> > >
> > >
> > > In terms of "However, analytical estimations using the methodology of [1,2] on 45nm CMOS is a gross estimation and really does not qualify as efficiency improvement. For e.g. you might be sparse, but if your hardware cannot take advantage of sparsity, you will in fact have more computations.": For the energy estimation, we followed the approach [1] as applied in Diet-SNN (IEEE TNNLS 2021) [2] and Dspike (NeuraIPS 2021) [3], which is a commonly recognized way for power consumption estimation of SNN. There could exist energy difference between analytical estimation and real measurement on neuromorphic hardware. However, **the argument of spike sparsity is not sufficient for power reduction does not mean it is not necessary**. A more accurate estimation of the power consumption of SNN depends on specific neuromorphic hardware and is beyond the scope of this work, so it should not influence our main contribution.
> > >
> > >
> > > **For other comments:**
> > >
> > >
> > > In terms of "I still think that with respect to previous works as the authors is marginal": **both AutoSNN and SNASNet are constrained to fixed network backbone and are limited to demonstrations on image classification**. In contrast, as stated in the general response, our approach **enables network level optimization of SNN and is effective in both classification and dense prediction tasks**. In addition, **we develop the DGS method which has been demonstrated of general usage for different SNN structures and robust to different SG functions** (in supplement experiments). We believe these contributions are not marginal.
> > >
> > >
> > > In terms of "The authors make an argument that their method does well on event-stereo tasks compared to ANNs. Again, there have been some recent work on neuromorphic datasets that show SOTA performance with SNNs.": To our best knowledge our work is the current SNN SOTA on the benchmark event-based stereo task of the MVSEC dataset. We would appreciate if the reviewer can provide more details of these mentioned recent works so we can compare.
> > >
> > >
> > > [1] M. Horowitz, “1.1 Computing’s energy problem (and what we can do about it),” in IEEE Int. Solid-State Circuits Conf. (ISSCC) Dig. Tech. Papers, Feb. 2014, pp. 10–14.
> > >
> > > [2] Nitin Rathi and Kaushik Roy. Diet-snn: A low-latency spiking neural network with direct input encoding and leakage and threshold optimization. IEEE Transactions on Neural Networks and Learning Systems, 2021.451
> > >
> > > [3] Yuhang Li, Yufei Guo, Shanghang Zhang, Shikuang Deng, Yongqing Hai, and Shi Gu. Differentiable spike: Rethinking gradient-descent for training spiking neural networks. Advances in Neural Information Processing Systems, 34, 2021.

---

### Author Response · Authors · 2022-08-01
**General response**

We thank all reviews for their time, constructive comments and valuable suggestions. While we will respond to each reviewer individually, we would like to first give a general response to the common issues and reemphasize our contributions to the field of SNN.


Reviewer Y1Po,cjsQ and LhUf have mentioned two exiting works related to NAS for SNN. Both AutoSNN [1] and SNASNet [2] were on arxiv ([1] has been recently accepted by ICML2022) and we didn’t notice them by the time we submitted to NeuraIPS. We acknowledge they are contemporary works and will cite them in revision. Specifically, [1] studied pooling operations for downsampling in SNNs and applied NAS to reduce the overall spike numbers of the network. [2] applied NAS to improve network initialization and explore backward connections. However, both works only searched for different SNN cells or combinations of them under fixed network backbone and their application is limited to image classification.


Our approach differs from both works in terms of method and application scope. Methodologically, our approach **optimize on both the cell and the architecture level of SNN using end-to-end differentiable hierarchical search**, which not only achieves SOTA accuracy on image classification of CIFAR (a comparison is provided in table below), but is also of **general usage and effective for hard dense prediction tasks** where architectures require more variation, **a field where few SNN works exist and significantly behind ANN performance ([3, 4, 5])**. Experimentally, **we demonstrate the superiority of SNN in processing sparse and dynamical signals in benchmark event-based stereo task**, in terms of accuracy, energy cost and inference speed, surpassing sophisticated designed ANN. In addition, we also extend the differentiable principle and **develop the DGS method to efficiently optimize SG functions and improve general SNN training**, as demonstrated in both classification and event-stereo task.


| Dataset      | Method | Architecture | Simulation Length | Params. [M] | Spikes[K] | Accuracy[%] |
| :---:        |    :----:   |          :---: | :---:        |    :----:   |      :---: |        :---: |
|CIFAR10      | SNASNet       |  SNASNet-Fw  | 5 | - | - | $93.12\pm 0.42$ |
|   |      |  SNASNet-Fw  | 8 | - | - | $93.64\pm 0.35$ |
|      |      |  SNASNet-Bw  | 5 | - | - | $93.73\pm 0.32$ |
|     |      |  SNASNet-Bw  | 8 | - | - | $94.12\pm 0.25$ |
|      | AutoSNN       |  AutoSNN (C=64)  | 8 | 5.44 | 261 | $92.54$ |
|      |        |  AutoSNN (C=128)  | 8 | 20.92 | 310 | $93.15$ |
|     |  DARTS-SNN   |  DARTS-SNN (n=4)  | 6 | 12.33 | 788 | $94.34\pm 0.06$ |
|     |  DARTS-SNN^D (1s)  |  DARTS-SNN (n=4)  | 6 | 12.33 | 865  | $94.68\pm 0.05$ |
|     |   DARTS-SNN |  DARTS-SNN (n=3)  | 6 | 14.29 | 752 | $95.35 \pm 0.05$ |
|     |   DARTS-SNN^D (1s)  |  DARTS-SNN (n=3)  | 6 | 14.29 | 724 | $95.36 \pm 0.01$ |
|     |   DARTS-SNN^D (5c) |  DARTS-SNN (n=3)  | 6 | 14.29 | 720 | $95.50 \pm 0.03$ |
|     |       |    |  |  |  |  |
|CIFAR100      | SNASNet       |  SNASNet-Fw  | 5 | - | - | $70.06\pm 0.45$ |
|      |      |  SNASNet-Bw  | 5 | - | - | $73.04\pm 0.36$ |
|      | AutoSNN       |  AutoSNN (C=64)  | 8 | - | - | $69.16$ |
|     |  DARTS-SNN   |  DARTS-SNN (n=4)  | 6 | 11.91 | 962 | $75.70\pm 0.14$ |
|     |  DARTS-SNN^D (1s)  |  DARTS-SNN (n=4)  | 6 | 11.91 | 1025 | $76.03\pm 0.20$ |


(-) means information not given. As shown in the table, our networks achieve higher accuracy compared to previous NAS-SNN works. The AutoSNN has fewer spikes due to specific optimization target. Furthermore, we made two changes in architecture in extended experiments and further improved the network performance. Specifically, we increased the output channel of the 1st stem to 144 (originally 108) and used 3 nodes (originally 4) within a cell. Instead of applying DGS to the first stem layer (1s), in new architecture we also applied DGS to the first node of the 5th cell (5c) which leads to our current best result.


[1] Na B, Mok J, Park S, et al. AutoSNN: Towards Energy-Efficient Spiking Neural Networks[J]. arXiv preprint arXiv:2201.12738, 2022.


[2] Kim Y, Li Y, Park H, et al. Neural architecture search for spiking neural networks[J]. arXiv preprint arXiv:2201.10355, 2022.


[3] Jesse Hagenaars, Federico Paredes-Vallés, and Guido De Croon. Self-supervised learning of event-based optical ﬂow with spiking neural networks. NeuraIPS 2021.


[4] Youngeun Kim, Joshua Chough, and Priyadarshini Panda. Beyond classifcation: Directly training spiking neural networks for semantic segmentation. arXiv preprint arXiv:2110.07742, 2021.


[5] Ulysse Rançon, Javier Cuadrado-Anibarro, Benoit R Cottereau, and Timothée Masquelier. Stereospike: Depth learning with a spiking neural network. arXiv preprint arXiv:2109.13751, 2021.

---

### Meta-Review · Area_Chair_vdcL · 2022-08-20

**Recommendation:** Accept
**Confidence:** Certain

**Metareview:**

This paper proposes a new architecture search algorithm for spiking neural networks (SNNs). The key insight is to optimize both the cell and the architecture level of the SNN. Convincing numerical results are provided on image classification tasks (CIFAR10, CIFAR100, and an event-based stereo task).

One concern raised by the reviewers regards the comparison to existing work (some of which appears to be very recent). This point is raised by all the four reviewers (although it has led to a rather large variance in their initial assessments). After an in-depth discussion between authors and reviewers and a discussion between AC and reviewers as well, it appears that this concern has been addressed in a satisfactory way. Other concerns (e.g., training pipeline and versatility by reviewer cjsQ) have been also resolved, and the remaining ones (measuring energy accurately as mentioned by reviewer LhUf, and computational overhead on neuromorphic hardware as mentioned by reviewer hUzC) have been regarded as out of scope.

In summary, the reviewers have found the authors’ response convincing and have reached a consensus towards accepting the paper. After my own reading of the manuscript, I agree with this assessment and I am happy to recommend acceptance. As a final note, I would like to encourage the authors to include in the camera ready the discussions related to the feedback from the reviewers.


**Award:**

No

---

### Decision · Program_Chairs · 2022-09-14

Accept